# Online Deep Equilibrium Learning for Regularization by Denoising

**Jiaming Liu**[*]
Washington University in St. Louis
`jiaming.liu@wustl.edu`

**Xiaojian Xu**[*]
Washington University in St. Louis
`xiaojianxu@wustl.edu`

**Weijie Gan**
Washington University in St. Louis
`weijie.gan@wustl.edu`

**Shirin Shoushtari**
Washington University in St. Louis
`s.shirin@wustl.edu`

**Ulugbek S. Kamilov**
Washington University in St. Louis
`kamilov@wustl.edu`

## Abstract

Plug-and-Play Priors (PnP) and Regularization by Denoising (RED) are widely-used frameworks for solving imaging inverse problems by computing fixed-points of operators combining physical measurement models and learned image priors. While traditional PnP/RED formulations have focused on priors specified using image denoisers, there is a growing interest in learning PnP/RED priors that are end-to-end optimal. The recent Deep Equilibrium Models (DEQ) framework has enabled memory-efficient end-to-end learning of PnP/RED priors by implicitly differentiating through the fixed-point equations without storing intermediate activation values. However, the dependence of the computational/memory complexity of the measurement models in PnP/RED on the total number of measurements leaves DEQ impractical for many imaging applications. We propose ODER as a new strategy for improving the efficiency of DEQ through stochastic approximations of the measurement models. We theoretically analyze ODER giving insights into its ability to approximate the traditional DEQ approach for solving inverse problems. Our numerical results suggest the potential improvements in training/testing complexity due to ODER on three distinct imaging applications.

## 1 Introduction

There has been considerable recent interest in using *deep learning (DL)* in the context of imaging inverse problems [1–8]. Instead of explicitly defining a regularizer, the traditional DL approach is based on training a *convolutional neural network (CNN)* architecture, such as U-Net [9], to invert the measurement operator by exploiting the natural redundancies in the imaging data [10–14]. *Plug-and-Play Priors (PnP)* [15] and *Regularization by Denoising (RED)* [16] are two well-known alternative approaches to the traditional DL that enable the integration of pre-trained CNN denoisers, such as DnCNN [17] or DRUNet [18], as image priors within iterative algorithms. When equipped with advanced CNN denoisers, PnP/RED provide excellent performance by exploiting both the implicit prior, characterized by a denoiser, and the measurement model [19–27]. *Deep Unfolding (DU)* is a related approach that interprets the iterations of an image recovery algorithm as layers of a neural

---

[*]These authors contributed equally.

36th Conference on Neural Information Processing Systems (NeurIPS 2022).

network and trains it end-to-end in a supervised fashion. Unlike in PnP/RED, the CNN in DU is trained jointly with the measurement model, leading to an image prior optimized for a given inverse problem [28–34]. DU architectures, however, are usually limited to a small number of unfolded iterations due to the high computational and memory complexity of training.

Recent work on *Neural ODEs* [35–37] and *Deep Equilibrium Models (DEQ)* [38–43] has shown the potential benefits of *implicit neural networks* in a number of DL tasks. For example, DEQ was recently used to train CNN priors within PnP/RED iterations by differentiating through the fixed points of the corresponding iterations [40]. Training PnP/RED using DEQ is equivalent to training an infinite depth feedforward network integrating a physical measurement model and CNN prior. However, the training of such networks can still be a significant computational and memory challenge in applications that require processing of a large number of sensor measurements. Specifically, the data-consistency layers in [40] are based on *batch* processing, which means that the *entire set of measurements* is processed at each layer. While this type of batch data processing is known to be suboptimal in traditional large-scale optimization [44–48], the issue has never been considered in the context of training of implicit networks such as those specified via PnP/RED iterations.

This paper addresses this issue by proposing *Online Deep Equilibrium RED (ODER)* as the first DEQ framework for inverse problems that adopts *stochastic processing* of measurements within an implicit neural network. We argue that the proposed *online* approach can improve training and testing efficiency compared to its *batch* counterpart in a number of applications where the number of measurements is large. ODER can be implemented using the fixed-point iterations of RED by introducing stochastic approximations to the corresponding forward and backward DEQ passes. The CNN prior within ODER is trained end-to-end to remove artifacts due to the imaging system and stochastic processing. Our theoretical analysis provides explicit error bounds on the training accuracy of *implicit online neural networks* used in ODER under a set of clearly specified assumptions. We show the practical relevance of ODER by solving inverse problems in *intensity diffraction tomography (IDT)* [26, 49], sparse-view *computed tomography (CT)* [50] and accelerated parallel *magnetic resonance imaging (MRI)* [51, 52]. Our numerical results show the ability of ODER to match the imaging quality of the batch DEQ learning at a fraction of complexity. Our work thus addresses an important gap in the current literature on PnP/RED, DU, and DEQ by providing an efficient framework applicable to a wide variety of imaging inverse problems.

All proofs and some technical details that have been omitted for space appear in the appendix, which also provides more background and simulations. The code for our numerical evaluation is available at: https://github.com/wustl-cig/ODER.

## 2   Background

**Inverse problems.** Many imaging problems—such as IDT, CT, and MRI—can be formulated as an inverse problem involving the recovery of an image $\boldsymbol{x}^* \in \mathbb{R}^n$ from noisy measurements $\boldsymbol{y} = \boldsymbol{A}\boldsymbol{x}^* + \boldsymbol{e}$, where $\boldsymbol{A} \in \mathbb{R}^{m \times n}$ is the measurement operator and $\boldsymbol{e} \in \mathbb{R}^m$ is the noise. A common approach to estimate $\boldsymbol{x}^*$ is to solve an optimization problem

$$\widehat{\boldsymbol{x}} = \arg\min_{\boldsymbol{x} \in \mathbb{R}^n} \left\{ g(\boldsymbol{x}) + h(\boldsymbol{x}) \right\}, \tag{1}$$

where $g$ is a data-fidelity term that quantifies consistency with the observed data $\boldsymbol{y}$ and $h$ is a regularizer that encodes prior knowledge on $\boldsymbol{x}$. A widely-used data-fidelity term and regularizer in inverse problems are $g(\boldsymbol{x}) = \frac{1}{2}\|\boldsymbol{y} - \boldsymbol{A}\boldsymbol{x}\|_2^2$ and the *total variation (TV)* function $h(\boldsymbol{x}) = \tau\|\boldsymbol{D}\boldsymbol{x}\|_1$, where $\boldsymbol{D}$ is the gradient operator and $\tau > 0$ is the regularization parameter [53–55].

**PnP, RED, and DU.** PnP [15, 20] and RED [16] are two related classes of iterative algorithms that use *additive white Gaussian noise (AWGN)* denoisers, such as BM3D [56] or DnCNN [17], as priors for inverse problems (see the recent review [57]). Since for general denoisers PnP/RED do not solve an optimization problem [23], it is common to interpret PnP/RED as fixed-point iterations of some high-dimensional operators. For example, given a denoiser $\mathsf{D}_{\boldsymbol{\theta}} : \mathbb{R}^n \to \mathbb{R}^n$ parameterized by a CNN with weights $\boldsymbol{\theta}$, the *steepest descent* variant of RED (SD-RED) [16] can be written as

$$\boldsymbol{x}^k = \mathsf{T}_{\boldsymbol{\theta}}(\boldsymbol{x}^{k-1}) = \boldsymbol{x}^{k-1} - \gamma\mathsf{G}_{\boldsymbol{\theta}}(\boldsymbol{x}^{k-1}) \quad \text{with} \quad \mathsf{G}_{\boldsymbol{\theta}}(\boldsymbol{x}) := \nabla g(\boldsymbol{x}) + \tau(\boldsymbol{x} - \mathsf{D}_{\boldsymbol{\theta}}(\boldsymbol{x})), \tag{2}$$

where $g$ is the data-fidelity term, and $\gamma, \tau > 0$ are the step size and the regularization parameters, respectively. SD-RED thus seeks to compute a fixed-point $\overline{\boldsymbol{x}} \in \mathbb{R}^n$ of the operator $\mathsf{T}$

$$\overline{\boldsymbol{x}} \in \mathsf{Fix}(\mathsf{T}_{\boldsymbol{\theta}}) := \left\{ \boldsymbol{x} \in \mathbb{R}^n : \mathsf{T}_{\boldsymbol{\theta}}(\boldsymbol{x}) = \boldsymbol{x} \right\} \quad \Leftrightarrow \quad \mathsf{G}_{\boldsymbol{\theta}}(\overline{\boldsymbol{x}}) = \nabla g(\overline{\boldsymbol{x}}) + \tau(\overline{\boldsymbol{x}} - \mathsf{D}_{\boldsymbol{\theta}}(\overline{\boldsymbol{x}})) = \boldsymbol{0}, \tag{3}$$

The solutions of (3) balance the requirements to be both data-consistent (via $\nabla g$) and noise-free (via $(\mathsf{I} - \mathsf{D}_{\boldsymbol{\theta}})$), which can be intuitively interpreted as finding an equilibrium between the physical measurement model and learned prior model. Remarkably, this heuristic of using denoisers not necessarily associated with any $h$ within an iterative algorithm exhibited great empirical success [25, 27, 58–66] and spurred a great deal of theoretical work on PnP/RED [19, 22–24, 67–74]. It is worth mentioning that there has been considerable effort in reducing the *test-time* computational/memory complexity of PnP/RED by designing online and stochastic PnP/RED algorithms [26, 68, 72, 75].

DU (also known as *algorithm unrolling*) is a DL paradigm that has gained popularity due to its ability to systematically connect iterative algorithms and deep neural network architectures (see reviews in [3, 76]). Many PnP/RED algorithms have been turned into DU architectures by parameterizing the operator $\mathsf{D}_{\boldsymbol{\theta}}$ as a CNN with weights $\boldsymbol{\theta}$, truncating the PnP/RED algorithm to a fixed number of iterations, and training the corresponding architecture end-to-end in a supervised fashion. Recent work has explored strategies for reducing the memory and computational complexity of training DU architectures [77, 78], However, a key bottleneck in DU training is the necessity to store the intermediate activation values required for computing the backpropagation updates, which fundamentally limits the number of unfolding layers one can practically use in large-scale applications.

**DEQ.** DEQ [38] is a recent method for training infinite-depth, weight-tied feedforward networks by analytically backpropagating through the fixed points using implicit differentiation. The DEQ output is specified implicitly as a fixed point of an operator $\mathsf{T}_{\boldsymbol{\theta}}$ parameterized by weights $\boldsymbol{\theta}$

$$\overline{\boldsymbol{x}} = \mathsf{T}_{\boldsymbol{\theta}}(\overline{\boldsymbol{x}}) \,. \tag{4}$$

The DEQ forward pass estimates $\overline{\boldsymbol{x}}$ in (4) by either running a fixed-point iteration or using an optimization algorithm. The DEQ backward pass produces gradients with respect to $\boldsymbol{\theta}$ by implicitly differentiating through the fixed points without the knowledge of how they are estimated

$$\ell(\boldsymbol{\theta}) = \frac{1}{2}\|\overline{\boldsymbol{x}}(\boldsymbol{\theta}) - \boldsymbol{x}^*\|_2^2 \quad \Rightarrow \quad \nabla\ell(\boldsymbol{\theta}) = (\nabla_{\boldsymbol{\theta}}\mathsf{T}_{\boldsymbol{\theta}}(\overline{\boldsymbol{x}}))^{\mathsf{T}} \, (\mathsf{I} - \nabla_{\boldsymbol{x}}\mathsf{T}_{\boldsymbol{\theta}}(\overline{\boldsymbol{x}}))^{-\mathsf{T}} \, (\overline{\boldsymbol{x}} - \boldsymbol{x}^*), \tag{5}$$

where $\ell$ is the loss function, $\boldsymbol{x}^*$ is the training label, and $\mathsf{I}$ is the identity mapping. The vector product with the inverse-Jacobian in (5) can be approximated by solving the following fixed-point equation

$$\overline{\boldsymbol{b}} := (\mathsf{I} - \nabla_{\boldsymbol{x}}\mathsf{T}_{\boldsymbol{\theta}}(\overline{\boldsymbol{x}}))^{-\mathsf{T}} \, (\overline{\boldsymbol{x}} - \boldsymbol{x}^*) \quad \Rightarrow \quad \overline{\boldsymbol{b}} = (\nabla_{\boldsymbol{x}}\mathsf{T}_{\boldsymbol{\theta}}(\overline{\boldsymbol{x}}))^{\mathsf{T}} \, \overline{\boldsymbol{b}} + (\overline{\boldsymbol{x}} - \boldsymbol{x}^*) \,. \tag{6}$$

Recent work has also explored Jacobian-free DEQ by replacing the inverse-Jacobian with an identity mapping $\mathsf{I}$, leading to a faster training [42].

The comparison of equations (2), (3), and (4) highlights an elegant connection between PnP/RED and DEQ. This connection was explored in the recent work [40] by using DEQ for learning the weights of the CNN prior $\mathsf{D}_{\boldsymbol{\theta}}$ end-to-end within PnP/RED iterations. Within the framework of [40], PnP/RED is used for the forward pass and a backward pass is obtained by using (6) on the PnP/RED operators. Specifically, the CNN prior in SD-RED can be trained by running the backward pass using $\mathsf{T}_{\boldsymbol{\theta}}$ in (2)

$$\boldsymbol{b}^k = \mathsf{F}(\boldsymbol{b}^{k-1}) = (\nabla_{\boldsymbol{x}}\mathsf{T}_{\boldsymbol{\theta}}(\overline{\boldsymbol{x}}))^{\mathsf{T}} \, \boldsymbol{b}^{k-1} + (\overline{\boldsymbol{x}} - \boldsymbol{x}^*). \tag{7}$$

This work makes several new contributions to the existing literature on PnP/RED, DU, and DEQ. The focus is on *efficient training* of implicit networks by approximating $\mathsf{T}_{\boldsymbol{\theta}}$ in (4) with a "simpler" operator $\widehat{\mathsf{T}}_{\boldsymbol{\theta}}$. Following [40], we focus on inverse problems by using PnP/RED operators of form (2) that integrate the physical measurement models and learned CNN priors. We give algorithmic, theoretical, and numerical results that ODER leads to significant memory/computational gains, while preserving the performance of the original DEQ approach [40]. It is worth noting that the results here have the potential to generalize to many other implicit networks beyond those specified via PnP/RED.

## 3 Online Deep Equilibrium Method

We consider inverse problems where the data-fidelity term $g$ can be expressed as

$$g(\boldsymbol{x}) = \frac{1}{b}\sum_{i=1}^{b} g_i(\boldsymbol{x}), \tag{8}$$

where each $g_i$ depends only on the subset $\boldsymbol{y}_i \in \mathbb{R}^{m_i}$ of the full measurements $\boldsymbol{y} \in \mathbb{R}^m$ as

$$\mathbb{R}^m = \mathbb{R}^{m_1} \times \mathbb{R}^{m_2} \times \cdots \times \mathbb{R}^{m_b} \quad \text{with} \quad m = m_1 + m_2 + \cdots + m_b \,.$$

We are primarily interested in scenarios where the memory/computational complexity of the gradient $\nabla g$ is proportional to $b$. Thus, when $b \to \infty$, the memory and computational complexity of traditional DEQ to train the CNN prior within the batch PnP/RED algorithms becomes impractical.

To decouple the computational/memory complexity of DEQ from $b$, we adopt *online* processing of measurements, where $g$ is approximated using a minibatch of $w \ll b$ measurements

$$\widehat{g}(\boldsymbol{x}) = \frac{1}{w}\sum_{s=1}^{w} g_{i_s}(\boldsymbol{x}) \quad \Rightarrow \quad \nabla\widehat{g}(\boldsymbol{x}) = \frac{1}{w}\sum_{s=1}^{w} \nabla g_{i_s}(\boldsymbol{x}) \quad \Rightarrow \quad \mathsf{H}\widehat{g}(\boldsymbol{x}) = \frac{1}{w}\sum_{s=1}^{w} \mathsf{H}g_{i_s}(\boldsymbol{x}) , \quad (9)$$

where $\{i_1, \ldots, i_w\}$ are i.i.d random variables selected uniformly from the set $\{1, \ldots, b\}$. Note that (9) directly implies the *unbiasedness* of the online gradient $\mathbb{E}\left[\nabla\widehat{g}(\boldsymbol{x})\right] = \nabla g(\boldsymbol{x})$ and Hessian $\mathbb{E}\left[\mathsf{H}\widehat{g}(\boldsymbol{x})\right] = \mathsf{H}g(\boldsymbol{x})$ with the expectations taken over the random indices $\{i_1, \ldots, i_w\}$.

### 3.1 ODER Learning

ODER seeks to minimize the MSE loss over $p \geq 1$ training samples

$$\ell(\boldsymbol{\theta}) = \frac{1}{p}\sum_{j=1}^{p} \ell_j(\boldsymbol{\theta}) \quad \text{with} \quad \ell_j(\boldsymbol{\theta}) = \frac{1}{2}\|\overline{\boldsymbol{x}}_j(\boldsymbol{\theta}) - \boldsymbol{x}_j^*\|_2^2 , \quad (10)$$

using *approximate* gradients $\nabla\widehat{\ell}_j(\boldsymbol{\theta})$ computed via the online forward and backward passes that are independent of $b$ (see Sections 3.2 and 3.3). Here, $\boldsymbol{\theta}$ denotes the weights of the CNN prior, $\boldsymbol{x}_j^*$ is the $j$th training label, and $\overline{\boldsymbol{x}}_j(\boldsymbol{\theta})$ is the fixed-point of the full-batch SD-RED algorithm (3). ODER can be trained using any gradient-based optimizer, such as the *stochastic gradient descent (SGD)*. At training iteration $t \geq 1$ we generate two sets of independent random variables. First, the index $j_t$ is selected uniformly at random from $\{1, \ldots, p\}$, then online forward and backward passes are computed using the measurement models in (9). We can thus express the SGD update rule for ODER as follows

$$\boldsymbol{\theta}^{t+1} = \boldsymbol{\theta}^t - \beta\nabla\widehat{\ell}_{j_t}(\boldsymbol{\theta}^t) \quad \text{with} \quad \nabla\widehat{\ell}_{j_t}(\boldsymbol{\theta}^t) = \left[\nabla_{\boldsymbol{\theta}}\widehat{\mathsf{T}}_{\boldsymbol{\theta}^t}(\boldsymbol{x}_{j_t}^K)\right]^{\mathsf{T}} \boldsymbol{b}_{j_t}^K, \quad (11)$$

where $\beta > 0$ is the SGD learning rate, $\boldsymbol{x}_{j_t}^K$ and $\boldsymbol{b}_{j_t}^K$ are the final iterates of the ODER online forward and backward passes at the training index $j_t$ after $K \geq 1$ iterations.

### 3.2 Online Forward Pass

The forward-pass of ODER is performed as follows

$$\boldsymbol{x}^k = \widehat{\mathsf{T}}_{\boldsymbol{\theta}}(\boldsymbol{x}^{k-1}) = \boldsymbol{x}^{k-1} - \gamma(\nabla\widehat{g}(\boldsymbol{x}^{k-1}) + \tau\mathsf{R}_{\boldsymbol{\theta}}(\boldsymbol{x}^{k-1})), \quad k = 1, 2, \ldots, K, \quad (12)$$

where $\mathsf{R}_{\boldsymbol{\theta}} = \mathsf{I} - \mathsf{D}_{\boldsymbol{\theta}}$ is the residual of the CNN prior $\mathsf{D}_{\boldsymbol{\theta}}$. The residual $\mathsf{R}_{\boldsymbol{\theta}}$ takes artifact-corrupted images at the input and produces the corresponding artifacts at the output. Note how the ODER forward-pass is independent of $b$ since it uses a minibatch approximation $\nabla\widehat{g}$ in (9).

It is worth mentioning that when considered separately from ODER, the forward pass corresponds to the existing online RED algorithm [26,79]. The contribution of this work is thus not the forward pass, but its integration into the training of an implicit online neural network, resulting in a more scalable and flexible DEQ framework for inverse problems.

### 3.3 Online Backward Pass

The backward pass of ODER for the MSE loss is performed as follows

$$\boldsymbol{b}^k = \widehat{\mathsf{F}}(\boldsymbol{b}^{k-1}) = \left[\nabla_{\boldsymbol{x}}\widehat{\mathsf{T}}_{\boldsymbol{\theta}}\left(\boldsymbol{x}^K\right)\right]^{\mathsf{T}} \boldsymbol{b}^{k-1} + (\boldsymbol{x}^K - \boldsymbol{x}^*), \quad k = 1, 2, \ldots, K, \quad (13)$$

starting from $\boldsymbol{b}^0 = \boldsymbol{0}$, where $\boldsymbol{x}^K$ is the final iterate of the forward pass (12) at iteration $K \geq 1$. In both traditional and online backward passes, conventional auto-differentiation tools enable the computation of the Jacobian-vector products in (7) and (13). However, the key difference is that the computational complexity of ODER does not depend on the total number of measurements $b$.

# 4 Theoretical Analysis

Our main theoretical result in this section relies on a set of explicit assumptions and two propositions analyzing the online forward and backward passes. All the proofs will be provided in the supplement.

**Assumption 1.** *Each $g_i$ is twice continuously differentiable and convex. There exists $\lambda > 0$ such that each gradient $\nabla g_i$ and Hessian $\mathsf{H}g_i$ are $\lambda$-Lipschitz continuous.*

The fact that $g$ is twice continuously differentiable is needed for the backward pass. The assumption that all the Lipschitz constants are the same is only needed to streamline mathematical exposition.

**Assumption 2.** $\mathsf{D}_{\boldsymbol{\theta}}(\boldsymbol{x})$ *is continuously differentiable with respect to $\boldsymbol{\theta}$ and $\boldsymbol{x}$. There exists $\alpha > 0$ such that $\mathsf{D}_{\boldsymbol{\theta}}(\boldsymbol{x})$, $\nabla_{\boldsymbol{x}}\mathsf{D}_{\boldsymbol{\theta}}(\boldsymbol{x})$, and $\nabla_{\boldsymbol{\theta}}\mathsf{D}_{\boldsymbol{\theta}}(\boldsymbol{x})$ are $\alpha$-Lipschitz continuous with respect to $\boldsymbol{\theta}$ and $\boldsymbol{x}$. Finally, we also assume that $\mathsf{D}_{\boldsymbol{\theta}}$ is a contraction, which means that there exists $\kappa < 1$ such that*

$$\|\mathsf{D}_{\boldsymbol{\theta}}(\boldsymbol{z}) - \mathsf{D}_{\boldsymbol{\theta}}(\boldsymbol{y})\|_2 \leq \kappa\|\boldsymbol{z} - \boldsymbol{y}\|_2, \quad \forall \boldsymbol{z}, \boldsymbol{y} \in \mathbb{R}^n.$$

Since $\mathsf{D}_{\boldsymbol{\theta}}$ is a CNN, its differentiability is a standard assumption. The contractive $\mathsf{D}_{\boldsymbol{\theta}}$ and convex $g$, ensure that $\mathsf{T}_{\boldsymbol{\theta}}$ is a contraction, enabling provable convergence of the forward and backward passes. The design of contractive $\mathsf{T}_{\boldsymbol{\theta}}$ is a common PnP/RED strategy to ensure convergence [24, 40, 72].

**Assumption 3.** *There exists $R > 0$ such that for all $\overline{\boldsymbol{x}} \in \mathsf{Fix}(\mathsf{T})$ and $\overline{\boldsymbol{b}} \in \mathsf{Fix}(\mathsf{F})$, we have $\|\boldsymbol{x}^k - \overline{\boldsymbol{x}}\|_2 \leq R$ and $\|\boldsymbol{b}^k - \overline{\boldsymbol{b}}\|_2 \leq R$ for all $k \in \{1, \dots, K\}$.*

The existence of the bound $R$ is reasonable, as many images have bounded pixel values. Similarly, the bound on $\boldsymbol{b}^k$ is also reasonable for ensuring bounded DEQ gradients.

**Assumption 4.** *There exists $\nu > 0$ such that for all $\boldsymbol{x} \in \mathbb{R}^n$, we have*

$$\mathbb{E}\left[\|\nabla g(\boldsymbol{x}) - \nabla\widehat{g}(\boldsymbol{x})\|_2^2\right] \leq \frac{\nu^2}{w} \quad and \quad \mathbb{E}\left[\|\mathsf{H}g(\boldsymbol{x}) - \mathsf{H}\widehat{g}(\boldsymbol{x})\|_2^2\right] \leq \frac{\nu^2}{w},$$

*where the expectations are taken over $\{i_1, \dots, i_w\}$.*

The variance bounds are standard in stochastic algorithms. The variance bounds on the gradient and Hessian approximations are thus reasonable in this context. The decrease of the bounds for higher values of $w$ is natural since $\widehat{g}$ is an unbiased estimator of $g$ obtained averaging $w$ independent terms.

**Proposition 1.** *Run the forward pass of ODER for $k \geq 1$ iterations under Assumptions 1-4 using the step size $0 < \gamma < 1/(\lambda + \tau)$. Then, the sequence of forward pass iterates satisfies*

$$\mathbb{E}\left[\|\boldsymbol{x}^k - \overline{\boldsymbol{x}}\|_2\right] \leq \eta^k R + \frac{\gamma\nu}{(1-\eta)\sqrt{w}}, \tag{14}$$

*for some constant $0 < \eta < 1$ where $\overline{\boldsymbol{x}} \in \mathsf{Fix}(\mathsf{T})$.*

Proposition 1 is a variation on the convergence results for online RED/PnP [26, 68, 72, 75], showing that the forward pass converges to $\overline{\boldsymbol{x}} \in \mathsf{Fix}(\mathsf{T})$ up to an error term that can be controlled via $\gamma$ and $w$.

**Proposition 2.** *Run the backward pass of ODER for $k \geq 1$ iterations under Assumptions 1-4 from $\boldsymbol{b}^0 = \mathbf{0}$ using the step-size $0 < \gamma < 1/(\lambda + \tau)$. Then, the sequence of backward pass iterates satisfies*

$$\mathbb{E}\left[\|\boldsymbol{b}^k - \overline{\boldsymbol{b}}\|_2\right] \leq B_1\eta^k + \frac{B_2}{\sqrt{w}}, \tag{15}$$

*where $0 < \eta < 1$, $B_1 > 0$ and $B_2 > 0$ are constants independent of $k$ and $w$, and $\overline{\boldsymbol{b}} \in \mathsf{Fix}(\mathsf{F})$.*

Proposition 2 shows that the online backward pass in expectation converges to $\overline{\boldsymbol{b}}$ up to an error term that can be controlled via $w$. The complete expressions for constants $B_1$ and $B_2$ are in the proof.

**Assumption 5.** *Function $\ell$ has a global minimizer $\boldsymbol{\theta}^*$ and has a $L$-Lipschitz continuous gradient, which means that for all $\boldsymbol{\theta}, \boldsymbol{\phi}$, we have $\|\nabla\ell(\boldsymbol{\theta}) - \nabla\ell(\boldsymbol{\phi})\|_2 \leq L\|\boldsymbol{\theta} - \boldsymbol{\phi}\|_2$.*

**Assumption 6.** *The loss function in (10) and indices $\{j_t\}$ in (11) are such that*

$$\mathbb{E}\left[\nabla\ell_{j_t}(\boldsymbol{\theta})\right] = \nabla\ell(\boldsymbol{\theta}) \quad and \quad \mathbb{E}\left[\|\nabla\ell_{j_t}(\boldsymbol{\theta}) - \nabla\ell(\boldsymbol{\theta})\|_2^2\right] \leq \sigma^2,$$

*where the expectations are taken with respect to the random index uniformly as $j_t \in \{1, \dots, p\}$.*

The existence of a minimizer and the Lipschitz continuity of the loss gradient are standard assumptions in the literature [47,80,81]. Note that we do *not* assume that the training loss $\ell$ is convex. Assumption 6 is the standard assumption used in the analysis of SGD. We are now ready to state the main result.

**Main Theorem.** *Train ODER using SGD for $T \geq 1$ iterations under Assumptions 1-6 using the step-size parameters $0 < \beta \leq 1/L$ and the minibatch size $w \geq 1$. Select a large enough number of forward and backward pass iterations $K \geq 1$ to satisfy $0 < \eta^K \leq 1/\sqrt{w}$. Then, we have that*

$$\frac{1}{T} \sum_{t=0}^{T-1} \mathbb{E}\left[\|\nabla \ell(\boldsymbol{\theta}^t)\|_2^2\right] \leq \frac{2(\ell(\boldsymbol{\theta}^0) - \ell(\boldsymbol{\theta}^*))}{\beta T} + \frac{C_1}{\sqrt{w}} + \beta C_2 .$$

*where $C_1 > 0$ and $C_2 > 0$ are constants independent of $T$ and $w$.*

The expressions for constants $C_1$ and $C_2$ are in the proof. The theorem provides an explicit error bound on the iterates generated using (11) to approximate the stationary points of the desired loss (10). The error terms in the bound depend on the training step-size $\beta$ and the minibatch size $w$, both of which can be controlled during training. It is worth mentioning that our theoretical analysis of ODER is the first result in the literature that provides explicit bounds on learning *implicit online networks*.

## 5 Numerical Evaluation

We numerically validate ODER in the context of three computational imaging modalities: IDT, sparse-view CT, and parallel MRI. Our goal is to both (a) empirically evaluate the performance of ODER and (b) highlight its effectiveness for processing a large number of measurements. We adopt $\ell_2$-norm loss $g(\boldsymbol{x}) = \frac{1}{2}\|\boldsymbol{y} - \boldsymbol{A}\boldsymbol{x}\|_2^2$ as the data-fidelity term for all three imaging modalities.

ODER is compatible with any CNN architecture used to implement $\mathsf{D}_{\boldsymbol{\theta}}$. We use a *tiny* U-Net architecture [78] for ODER and the traditional RED (DEQ) [40]. We have added spectral normalization [82] to all the layers of CNN for stability (see the supplement for the numerical evaluation of the contractiveness of $\mathsf{T}_{\boldsymbol{\theta}}$ on all three modalities). Similar to [40], the CNN prior of ODER and RED (DEQ) are initialized using pre-trained denoisers. During the training of both ODER and RED (DEQ), we use the *Nesterov acceleration* [80] for the forward pass and *Anderson acceleration* [83] for the backward pass. We also adopt the stopping criterion from [40, 84] by setting residual tolerance to $10^{-3}$ for both forward and backward iterations (see supplement for additional details).

For reference we include several other well-known baseline methods, including TV [53], U-Net [9] and ISTA-Net$^+$ [28]. We also include the unfolded *RED (Unfold)* [78] and the traditional *RED (Denoising)* [16] to illustrate the improvements due to DEQ. TV is an iterative method that does not require training, while other methods are all DL-based with publicly available implementations. We use the U-Net architecture in [9] as the AWGN denoiser for RED, while we use the same tiny U-Net for RED (Unfold) as in RED (DEQ). For each imaging modality, we trained the denoiser in RED (Denoising) for AWGN removal at five noise levels corresponding to $\sigma \in \{2, 5, 7, 10, 15\}$. For each experiment, we select the denoiser achieving the highest SNR. In all the experiments, we train ODER and RED (DEQ) using the same training strategy and parameter initialization settings. We use `fminbound` in the `scipy.optimize` toolbox to identify the optimal regularization parameters for TV, RED (Denoising), ODER and RED (DEQ) at the inference time.

### 5.1 Image Reconstruction in IDT

IDT [49] is a data intensive computational imaging modality that seeks to recover the spatial distribution of the complex-valued permittivity contrast of an object given a set of its intensity-only measurements. Specifically, $\boldsymbol{A}$ consists of a set of $b$ complex measurement operators $[\boldsymbol{A}_1, \ldots, \boldsymbol{A}_b]^\mathsf{T}$, where each $\boldsymbol{A}_i$ is a convolution corresponding to the $i$th measurement $\boldsymbol{y}_i$. In the simulation, we randomly extracted and cropped 400 slices of $416 \times 416$ images for training, 28 images for validation and 56 images for testing from Brecahad database [85]. Following the setup in [26, 49], we generated $b = 500$ intensity measurements under AWGN corresponding to $\{15, 20, 25\}$ dB of input SNR. ODER and RED (DEQ) were trained at the noise level corresponding to 20 dB input SNR. In our comparisons, we also included the recent SGD-Net [78] method that corresponds to RED (Unfold), but uses stochastic data-consistency layers similar to ODER. SGD-Net allows for more unfolded iteration blocks by improving the usage of limited GPU memory. Both ODER and RED (DEQ) were trained using SGD, while all other methods were trained using Adam [86].

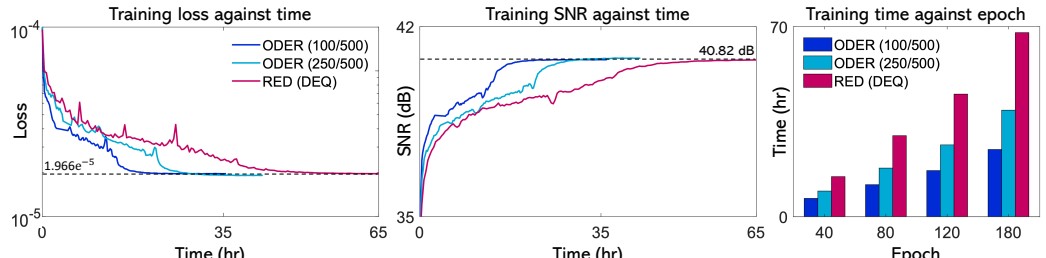

Figure 1: *Quantitative evaluation of ODER on IDT for two minibatch sizes $w \in \{100, 250\}$ used at each step of the network against RED (DEQ) using the full batch of $b = 500$ measurements. The left figure plots the loss against time in hours for different values of $w$ evaluated on the training set. The middle and right figures plot the corresponding SNR against time and the amount of time required to reach a certain epoch for different values of $w$. By using minibatches $1 \leq w \leq b$, ODER can achieve nearly $2.5\times$ improvement in training time over RED (DEQ) for a similar final imaging quality.*

Table 1: IDT image recovery for different input SNR (dB) values on images from [85]. We also present model size and per-iteration memory usage for the measurements, and average test-times.

| Method | Input SNR (dB) | | | Size | | Time | |
|---|---|---|---|---|---|---|---|
| | 15 | 20 | 25 | Model | Meas. | CPU | GPU |
| **TV** | 38.34 | 38.77 | 38.85 | —— | 3.56 GB | 215.3s | 32.24s |
| **U-Net** | 38.35 | 38.89 | 39.02 | 118.2 MB | —— | 2.811s | 0.089s |
| **ISTA-Net+** | 38.37 | 38.94 | 39.27 | 1.21 MB | 3.56 GB | 7.081s | 0.216s |
| **SGD-Net (100)** | 39.62 | 40.26 | 40.47 | 29.7 MB | 0.71GB | 6.697s | 0.207s |
| **RED (Denoising)** | 39.52 | 40.04 | 40.41 | 118.2 MB | 3.56 GB | 285.5s | 7.528s |
| **ODER (100)** | 40.28 | 41.42 | 41.94 | 29.7 MB | 0.71 GB | 63.31s | 2.051s |
| **ODER (250)** | **40.57** | 41.50 | **41.96** | 29.7 MB | 1.76 GB | 118.7s | 3.628s |
| **RED (DEQ)** | 40.54 | **41.51** | 41.95 | 29.7 MB | 3.56 GB | 202.3s | 6.362s |

Fig. 1 compares the average loss and SNR achieved by RED (DEQ) with ($b = 500$) and ODER with $w \in \{100, 250\}$ during training. It took 67.49 hours to train RED (DEQ) for 180 epochs. It took 24.76 and 39.23 hours to train ODER with ($w = 100$) and ($w = 250$), respectively, for the same number epochs. Table 1 provides the final SNR achieved by ODER and several baseline methods on the test data. The runtime in the table corresponds to the average inference time that excludes the model loading. ODER with ($w = 100$) is around $3\times$ faster than RED (DEQ) on both GPU and CPU. Fig. 2 (*left*) highlights the faster convergence of ODER compared to RED (DEQ) to the similar SNR.

ODER is memory efficient due to its online processing of measurements. The memory considerations in IDT include the size of all the variables related to the desired image $x$, the measured data $y_i$, and the variables related to the measurement operator $\{A_i\}$. ODER addresses the problem of storing and processing the measurements and the measurement operators on the GPU during end-to-end training. Table 1 shows the total memory (GB) used by ODER and RED (DEQ) for reconstructing a $416 \times 416$ pixel permittivity image. While RED (DEQ) requires 3.56 GB of GPU memory in every iteration, ODER with $w = 100$ requires only 0.71 GB, which is about $20\%$ of the full volume.

## 5.2  Image Reconstruction in Sparse-View CT

We consider simulated data obtained from the clinically realistic CT images provided by Mayo Clinic for the *low dose CT grand challenge* [87]. Specifically, 2070 2D slices of size $512 \times 512$ corresponding to 7 patients were used to train the models. The test images correspond to 55 slices randomly selected from another patient. We implement the measurement operator $A$ and its adjoint $A^{\mathsf{T}}$ with PyTorch implementation of `Radon` and `IRadon`[2] transform. We assume that the CT machine is designed to project from nominal angles with $b \in \{90, 120, 180\}$ projection views that are evenly-distributed on a half circle and 724 detector pixels. We add Gaussian noise to the sinograms to make the input SNR equal to 50 dB. We empirically found that using Adam [86] is around $2\times$ faster than applying SGD when training both ODER and RED (DEQ). We thus trained all learning-based

---
[2]The code is publicly available at `https://github.com/phernst/pytorch_radon`

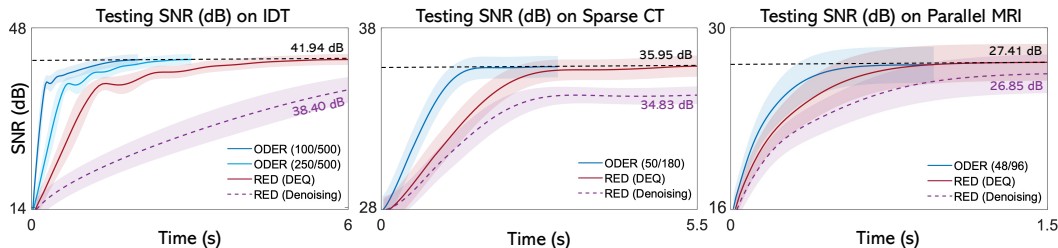

Figure 2: *Illustration of the convergence speed of ODER, RED (DEQ) and RED (Denoising) for three imaging applications.* **Left:** *IDT with the full batch of $b = 500$ measurements under 25 dB input SNR.* **Middle:** *Sparse-view CT with $b = 180$ projection views.* **Right:** *Parallel MRI at $20\%$ sampling with $b = 96$ simulated coil sensitivity maps. ODER achieves $1.4\times \sim 3\times$ speedup over RED (DEQ) at inference time without significant degradation in accuracy across three problems.*

Table 2: Sparse-view CT image recovery in terms of SNR (dB) and SSIM on test images from [87]. The last two columns provide the average test-times for a $512 \times 512$ image using 180 views.

| Method | Projection Views | | | | | | Time | |
|---|---|---|---|---|---|---|---|---|
| | 90 | | 120 | | 180 | | CPU | GPU |
| **TV** | 29.44 | 0.9688 | 30.27 | 0.9731 | 31.33 | 0.9771 | 768.1s | 15.61s |
| **U-Net** | 33.05 | 0.9741 | 34.02 | 0.9790 | 35.11 | 0.9815 | 4.014s | 0.056s |
| **ISTA-Net+** | 32.15 | 0.9706 | 33.38 | 0.9755 | 34.83 | 0.9812 | 37.38s | 0.344s |
| **RED (Unfold)** | 33.97 | 0.9753 | 35.01 | 0.9824 | 35.78 | 0.9835 | 29.93s | 0.256s |
| **RED (Denoising)** | 32.64 | 0.9708 | 33.60 | 0.9789 | 34.83 | 0.9807 | 498.5s | 5.549s |
| **ODER** | 34.40 | 0.9824 | 35.12 | 0.9841 | 35.91 | 0.9859 | 334.1s | 3.113s |
| **RED (DEQ)** | **34.61** | **0.9826** | **35.26** | **0.9845** | **35.95** | **0.9861** | 616.1s | 5.466s |

methods using Adam. Table 2 reports the average SNR and SSIM results for ODER with ($w/b$) of $\{30/90, 40/120, 50/180\}$ and all baselines. Fig. 2 (middle) reports the convergence speed of ODER with ($w = 50$) for sparse-view CT with full batch ($b = 180$) views. The visual comparisons are in Fig. 3 (bottom). Note how ODER matches the performance of RED (DEQ) and outperforms RED (Denoising) and RED (Unfold) across different projection views.

## 5.3 Image Reconstruction in Accelerated Parallel MRI

We simulated a multi-coil CS-MRI setup using radial Fourier sampling [88, 89]. The measurement operator $\boldsymbol{A}$ thus consists of a set of $b$ complex measurement operators depending on a set of receiver coils $\{\boldsymbol{S}_i\}$ [90]. For each coil, we have $\boldsymbol{A}_i = \boldsymbol{PFS}_i$, where $\boldsymbol{P}$ is the diagonal sampling matrix, $\boldsymbol{F}$ is the Fourier transform, and $\boldsymbol{S}_i$ is the diagonal matrix of sensitivity maps. ODER is evaluated on two brain MRI datasets. The first dataset [28] provides 800 slices of $256 \times 256$ images for training and 50 slices for testing. The second dataset [91] contains a randomly selected 400 volumes of $320 \times 320 \times 10$ images for training, and 32 volumes for testing. We synthesized the total number of ($b = 96$) 2D/3D coil sensitivity maps using the SigPy [92] for each dataset, respectively. Since all the CNNs in our numerical study are 2D, we apply them slice-by-slice when forming 3D volumes (all slices are passed in parallel using batch processing). We trained all learning-based methods using Adam. Fig. 2 (right) reports the convergence speed of ODER for CS-MRI at $20\%$ sampling. Table 3 reports the average SNR and SSIM values for ODER with ($w = 48$) and all baseline methods. The visual comparison can be found in Fig. 3 (top) at $10\%$ sampling.

## 6 Conclusion and Future Work

This work proposes ODER as a new online DEQ learning method for RED, analyzes its theoretical properties in terms of convergence and accuracy, and applies it to three widely-used imaging inverse problems. ODER extends the recent DEQ approach in [40] by introducing randomized processing of measurements. Our extensive theoretical and numerical results corroborate the potential of ODER to reduce the computational/memory complexity of training and testing, while achieving similar

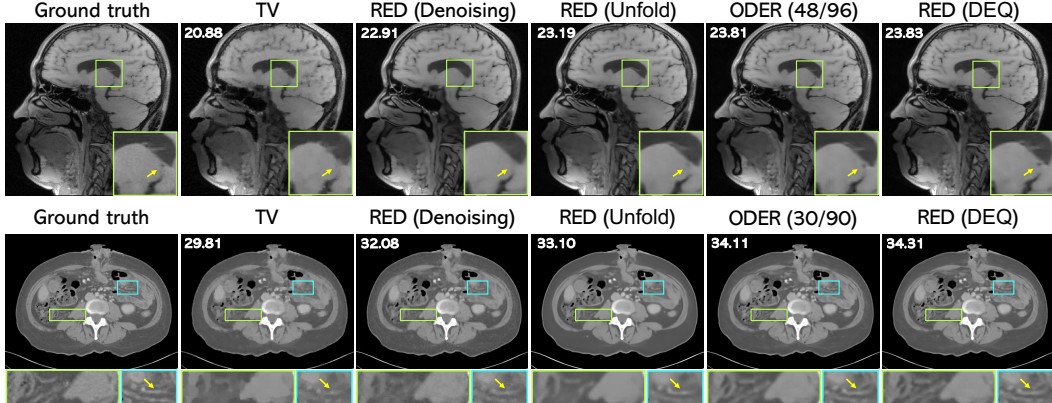

Figure 3: *Visual evaluation of several well-known methods on two imaging problems: (top) Reconstruction of a brain image from its radial Fourier measurements at $10\%$ sampling with $b = 96$ simulated coil sensitivity maps; (bottom) Reconstruction of a body CT image from $b = 90$ projection views. Note the similar performance of ODER and RED (DEQ), and the improvement over RED (Denoising) /RED (Unfold) due to the usage of DEQ learning. Best viewed by zooming in the display.*

Table 3: Average SNR (dB), SSIM, and running times for several methods on MRI images. The last two columns provide the average test-times for a $320 \times 320$ image using 96 simulated coils.

| Method | MRI Set1 [28] | | | | MRI Set2 [91] | | Time | |
|---|---|---|---|---|---|---|---|---|
| | 10% | | 20% | | 10% | | CPU | GPU |
| **TV** | 20.88 | 0.9059 | 24.87 | 0.9445 | 24.84 | 0.9674 | 122.2s | 7.591s |
| **U-Net** | 23.07 | 0.9329 | 26.42 | 0.9562 | 26.04 | 0.9712 | 0.683s | 0.011s |
| **ISTA-Net+** | 22.95 | 0.9298 | 26.31 | 0.9546 | 25.82 | 0.9693 | 8.993s | 0.264s |
| **RED (Unfold)** | 23.37 | 0.9363 | 26.81 | 0.9591 | 26.37 | 0.9744 | 8.744s | 0.231s |
| **RED (Denoising)** | 23.29 | 0.9352 | 26.85 | 0.9598 | 26.42 | 0.9748 | 272.4s | 7.511s |
| **ODER** | 24.08 | 0.9442 | 27.22 | 0.9649 | 27.03 | 0.9783 | 120.0s | 3.005s |
| **RED (DEQ)** | **24.10** | **0.9451** | **27.41** | **0.9660** | **27.10** | **0.9789** | 166.9s | 4.577s |

imaging quality as RED (DEQ). The future work can explore to further improve our analysis and design distributed variants of ODER to enhance its performance on parallel computing architectures.

# 7 Broader Impact

This work is expected to impact the area of imaging inverse problems with potential applications to computational microscopy, medical imaging, and image restoration. There is a growing need in imaging to deal with noisy and incomplete measurements by integrating multiple information sources, including physical sensor models and learned priors characterizing the properties of the desired image. The ability to accurately solve inverse problems might lead to new imaging tools for diagnosing health conditions, understanding biological processes, or inferring properties of complex materials. Learning based methods, including PnP/RED and ODER, have the potential to enable new technological capabilities; yet, they also come with a downside of being more complex, requiring high-levels of technical sophistication from potential users. While we aim to use our method to enable positive contributions to humanity, one can also imagine nonethical usage of imaging technology.

## Acknowledgments and Disclosure of Funding

Research presented in this article was supported in part by the NSF CAREER award CCF-2043134.

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
