# Supplementary Material for "Online Deep Equilibrium Learning for Regularization by Denoising"

**Jiaming Liu**[*]
Washington University in St. Louis
jiaming.liu@wustl.edu

**Xiaojian Xu**[*]
Washington University in St. Louis
xiaojianxu@wustl.edu

**Weijie Gan**
Washington University in St. Louis
weijie.gan@wustl.edu

**Shirin Shoushtari**
Washington University in St. Louis
s.shirin@wustl.edu

**Ulugbek S. Kamilov**
Washington University in St. Louis
kamilov@wustl.edu

## Abstract

Plug-and-Play Priors (PnP) and Regularization by Denoising (RED) are widely-used frameworks for solving imaging inverse problems by computing fixed-points of operators combining physical measurement models and learned image priors. While traditional PnP/RED formulations have focused on priors specified using image denoisers, there is a growing interest in learning PnP/RED priors that are end-to-end optimal. The recent Deep Equilibrium Models (DEQ) framework has enabled memory-efficient end-to-end learning of PnP/RED priors by implicitly differentiating through the fixed-point equations without storing intermediate activation values. However, the dependence of the computational/memory complexity of the measurement models in PnP/RED on the total number of measurements leaves DEQ impractical for many imaging applications. We propose ODER as a new strategy for improving the efficiency of DEQ through stochastic approximations of the measurement models. We theoretically analyze ODER giving insights into its convergence and ability to approximate the traditional DEQ approach. Our numerical results suggest the potential improvements in training/testing complexity due to ODER on three distinct imaging applications.

We adopt the monotone operator theory [1, 2] for a unified analysis of ODER. The contributions of this work are algorithmic, theoretical, and numerical. We propose ODER as a new algorithm. We then develop new theoretical insights into its ability to approximate the traditional DEQ. In Supplement A, we prove the convergence of forward pass to $\overline{x} \in \mathsf{Fix}(\mathsf{T})$ up to an error term controlled by $\gamma$ and $w$. In Supplement B, we prove that the online backward pass in expectation converges to $\overline{b} \in \mathsf{Fix}(\mathsf{F})$ up to an error term that can be controlled via $w$. In Supplement C, we prove our main theorem establishing the ability of ODER to approximate the stationary points of the desired loss $\ell(\boldsymbol{\theta})$ up to an error term that can be controlled *during training*. Finally, in Section D, we provide additional technical details on our implementations and simulations omitted from the main paper due to space.

We use the same notations as in the main paper. The measurement model corresponds to $\boldsymbol{y} = \boldsymbol{A}\boldsymbol{x}^* + \boldsymbol{e}$, where $\boldsymbol{x}^*$ is the true solution and $\boldsymbol{e}$ is the noise. The function $g(\boldsymbol{x})$ denotes the data-fidelity term. The operator $\mathsf{D}_{\boldsymbol{\theta}}(\cdot)$ denotes the learned prior within ODER and RED (DEQ), which is implemented via its residual $\mathsf{R}_{\boldsymbol{\theta}} := \mathsf{I} - \mathsf{D}_{\boldsymbol{\theta}}$. The operator $\widehat{\mathsf{T}}_{\boldsymbol{\theta}}(\cdot)$ and $\widehat{\mathsf{F}}(\cdot)$ denote the ODER stochastic forward and backward passes, respectively. The operator $\mathsf{T}_{\boldsymbol{\theta}}(\cdot)$ and $\mathsf{F}(\cdot)$ denote the full batch forward and

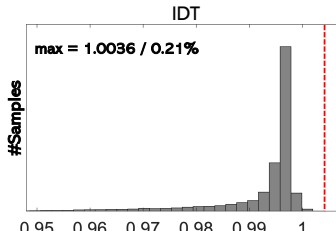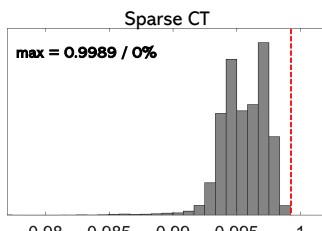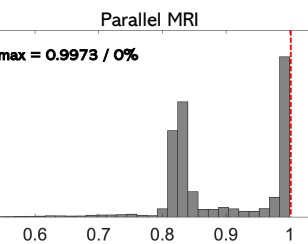

Figure 1: *Empirical evaluation of the Lipschitz continuity of* T. *Each histogram was generated by storing all ODER iterates and* $\overline{x} \in \mathsf{Fix}(\mathsf{T})$ *across all the test images used in the tables of the main paper. The x-axis is the value of* $\|\mathsf{T}(x^{k-1}) - \mathsf{T}(\overline{x})\|_2 / \|x^{k-1} - \overline{x}\|_2$. *Left: The histogram of IDT at* $b = 500$ *with {15, 20, 25} dB of input SNR. Middle: The histogram of sparse CT at* $b \in \{90, 120, 180\}$ *projection views. Right: The histogram of the radially sub-sampled parallel MRI at* $10\%$ *and* $20\%$ *sampling. Note how* T *numerically acts as a contraction on all the iterates generated for the CT and MRI experiments, and on 99.79% iterates generated for the IDT experiments. Despite their imperfect numerical precision, current spectral normalization techniques still provide a powerful tool for systematically ensuring stability of PnP/RED fixed-point iterations.*

backward passes of RED (DEQ), respectively. Finally, our code, including pre-trained CNN models used in ODER and RED (DEQ), is also included in the supplementary material.

## A    Proof of Proposition 1

The following proposition shows the convergence (in expectation) of the ODER forward pass for convex $g$ and contractive $\mathsf{D}_{\theta}$. Note that this proof is a variation of existing results in the literature on online PnP/RED [3–6]. However, this result plays an important role in the analysis of the ability of ODER to approximate the traditional DEQ learning.

**Proposition 1.** *Run the forward pass of ODER for* $k \geq 1$ *iterations under Assumptions 1-4 using the step size* $0 < \gamma < 1/(\lambda + \tau)$. *Then, the sequence of forward pass iterates satisfies*

$$\mathbb{E}\left[\|x^k - \overline{x}\|_2\right] \leq \eta^k R + \frac{\gamma \nu}{(1 - \eta)\sqrt{w}}, \tag{1}$$

*for some constant* $0 < \eta < 1$ *where* $\overline{x} \in \mathsf{Fix}(\mathsf{T})$.

*Proof.* For notation connivance, we abbreviate $\mathsf{T}_{\theta}(\cdots)$ as $\mathsf{T}_{\theta}(\cdots)$ in the following proof. From Lemma 1, $\mathsf{T}$ is a contraction, which means that there exists $0 < \eta < 1$ such that

$$\|\mathsf{T}(z) - \mathsf{T}(y)\|_2 \leq \eta \|z - y\|_2,$$

for all $z, y \in \mathbb{R}^n$. Then, for $\overline{x} \in \mathsf{Fix}(\mathsf{T})$, we have that

$$\|x^k - \overline{x}\|_2 = \|\widehat{\mathsf{T}}(x^{k-1}) - \mathsf{T}(\overline{x})\|_2^2 = \|\mathsf{T}(x^{k-1}) - \mathsf{T}(\overline{x}) + \widehat{\mathsf{T}}(x^{k-1}) - \mathsf{T}(x^{k-1})\|_2$$

$$\leq \|\mathsf{T}(x^{k-1}) - \mathsf{T}(\overline{x})\|_2 + \gamma \|\mathsf{T}(x^{k-1}) - \widehat{\mathsf{T}}(x^{k-1})\|_2$$

$$\leq \eta \|x^{k-1} - \overline{x}\|_2 + \|\nabla g(x^{k-1}) - \nabla \widehat{g}(x^{k-1})\|_2,$$

where we used the triangular inequality and that $\mathsf{T}$ is $\eta$-Lipschitz continuous. We take the conditional expectation from both sides to obtain

$$\mathbb{E}\left[\|x^k - \overline{x}\|_2 \mid x^{k-1}\right] \leq \eta \|x^{k-1} - \overline{x}\|_2 + \frac{\gamma \nu}{\sqrt{w}},$$

where we applied the Jensen's inequality to the variance bound in Assumption 4. By taking the total expectation, we thus have

$$\mathbb{E}\left[\|x^k - \overline{x}\|_2\right] \leq \eta \mathbb{E}\left[\|x^{k-1} - \overline{x}\|_2\right] + \frac{\gamma \nu}{\sqrt{w}}.$$

By iterating this inequality and using the bound in Assumption 3, we get the result

$$\mathbb{E}\left[\|x^k - \overline{x}\|_2\right] \leq \eta^k R + \frac{\gamma \nu}{(1 - \eta)\sqrt{w}}.$$

$\square$

### A.1 Useful Results for the Proof of Proposition 1

The following lemma establishes that $\mathsf{T}$ is a contraction. The proof is a minor modification of the Proposition 1 from [7], which we provide for completeness. It is worth noting that this result does not assume that the functions $\{g_i\}$ are strongly convex.

**Lemma 1.** *Suppose that Assumptions 1-2 in the main paper are true. Then, for any* $0 < \gamma < 1/(\lambda+\tau)$, *the operator* $\mathsf{T}$ *in eq. (2) of the main paper is a contraction, which means that for all* $\boldsymbol{x} \in \mathbb{R}^n$

$$\|\nabla_{\boldsymbol{x}}\mathsf{T}(\boldsymbol{x})\|_2 < 1,$$

*where* $\|\cdot\|_2$ *denotes the spectral norm.*

*Proof.* The Jacobian of the operator $\mathsf{T}$ with respect to $\boldsymbol{x}$ is given by

$$\nabla_{\boldsymbol{x}}\mathsf{T}(\boldsymbol{x}) = (1 - \gamma\tau)\boldsymbol{I} - \gamma\mathsf{H}g(\boldsymbol{x}) - \gamma\tau\nabla_{\boldsymbol{x}}\mathsf{D}(\boldsymbol{x}).$$

Let $\lambda_1 \geq \cdots \geq \lambda_n$ denote sorted eigenvalues of the Hessian matrix $\mathsf{H}g(\boldsymbol{x})$. Since $g$ is convex, we have that $\lambda_n \geq 0$. Then, for any $\boldsymbol{x} \in \mathbb{R}^n$, we have

$$\begin{aligned}
\|\nabla_{\boldsymbol{x}}\mathsf{T}(\boldsymbol{x})\|_2 &= \|(1 - \gamma\tau)\boldsymbol{I} - \gamma\mathsf{H}g(\boldsymbol{x}) - \gamma\tau\nabla_{\boldsymbol{x}}\mathsf{D}(\boldsymbol{x})\|_2 \\
&\leq \|(1 - \gamma\tau)\boldsymbol{I} - \gamma\mathsf{H}g(\boldsymbol{x})\|_2 + \gamma\tau\|\nabla_{\boldsymbol{x}}\mathsf{D}(\boldsymbol{x})\|_2 \\
&\leq \max_{1 \leq i \leq n}\{1 - \gamma\tau - \gamma\lambda_i\} + \gamma\tau\kappa \\
&\leq 1 - \gamma\tau(1 - \kappa) < 1,
\end{aligned}$$

where in the first inequality we used the triangular inequality, in the second the fact that $\mathsf{D}$ is a contraction, and in the third the convexity of $g$. $\qquad\square$

## B  Proof of Proposition 2

The following result is a novel analysis of the ODER backward pass. The result implies that the backward pass converges (in expectation) up to an error term that can be controlled by the minibatch parameter $w$. Our numerical results provide additional corroboration to our theory by showing that ODER nearly matches the performance of the traditional DEQ learning.

**Proposition 2.** *Run the backward pass of ODER for* $k \geq 1$ *iterations under Assumptions 1-4 from* $\boldsymbol{b}^0 = \boldsymbol{0}$ *using the step-size* $0 < \gamma < 1/(\lambda+\tau)$. *Then, the sequence of backward pass iterates satisfies*

$$\mathbb{E}\left[\|\boldsymbol{b}^k - \overline{\boldsymbol{b}}\|_2\right] \leq B_1\eta^k + \frac{B_2}{\sqrt{w}}, \tag{2}$$

*where* $0 < \eta < 1$, $B_1 > 0$ *and* $B_2 > 0$ *are constants independent of* $k$ *and* $w$, *and* $\overline{\boldsymbol{b}} \in \mathsf{Fix}(\mathsf{F})$.

*Proof.* Let $\boldsymbol{x}^K$ denote the output of the forward pass of ODER after $K \geq 1$ iterations, $\boldsymbol{x}^*$ denote the training label, and $\overline{\boldsymbol{x}} \in \mathsf{Fix}(\mathsf{T})$. Consider the following two operators

$$\mathsf{F}(\boldsymbol{b}) = [\nabla_{\boldsymbol{x}}\mathsf{T}(\overline{\boldsymbol{x}})]^{\mathsf{T}}\boldsymbol{b} + (\overline{\boldsymbol{x}} - \boldsymbol{x}^*) \quad \text{and} \quad \widehat{\mathsf{F}}(\boldsymbol{b}) = [\nabla_{\boldsymbol{x}}\widehat{\mathsf{T}}(\boldsymbol{x}^K)]^{\mathsf{T}}\boldsymbol{b} + (\boldsymbol{x}^K - \boldsymbol{x}^*),$$

where the first operator is used in the backward pass of RED (DEQ), while the second is its online approximation. Note also the following two Jacobians

$$\nabla_{\boldsymbol{x}}\mathsf{T}(\overline{\boldsymbol{x}}) = \boldsymbol{I} - \gamma(\mathsf{H}g(\overline{\boldsymbol{x}}) + \tau\nabla_{\boldsymbol{x}}\mathsf{R}(\overline{\boldsymbol{x}})) \quad \text{and} \quad \nabla_{\boldsymbol{x}}\widehat{\mathsf{T}}(\boldsymbol{x}^K) = \boldsymbol{I} - \gamma(\mathsf{H}\widehat{g}(\boldsymbol{x}^K) + \tau\nabla_{\boldsymbol{x}}\mathsf{R}(\boldsymbol{x}^K)).$$

Lemma 1 implies that $\mathsf{T}$ is a contraction. Let $0 < \eta < 1$ denote the Lipschitz constant of $\mathsf{T}$. Since $\nabla_{\boldsymbol{b}}\mathsf{F}(\boldsymbol{b}) = \nabla_{\boldsymbol{x}}\mathsf{T}(\overline{\boldsymbol{x}})$, we have $\|\nabla_{\boldsymbol{b}}\mathsf{F}(\boldsymbol{b})\|_2 = \|\nabla_{\boldsymbol{x}}\mathsf{T}(\overline{\boldsymbol{x}})\|_2 \leq \eta$, which means that $\mathsf{F}$ is a contraction

$$\|\mathsf{F}(\boldsymbol{z}) - \mathsf{F}(\boldsymbol{y})\|_2 \leq \eta\|\boldsymbol{z} - \boldsymbol{y}\|_2, \quad \boldsymbol{z}, \boldsymbol{y} \in \mathbb{R}^n.$$

We can thus show the following bound

$$\begin{aligned}
\|\boldsymbol{b}^k - \overline{\boldsymbol{b}}\|_2 &= \|\widehat{\mathsf{F}}(\boldsymbol{b}^{k-1}) - \mathsf{F}(\overline{\boldsymbol{b}})\|_2 = \|\mathsf{F}(\boldsymbol{b}^{k-1}) - \mathsf{F}(\overline{\boldsymbol{b}}) + \widehat{\mathsf{F}}(\boldsymbol{b}^{k-1}) - \mathsf{F}(\boldsymbol{b}^{k-1})\|_2 \\
&\leq \|\mathsf{F}(\boldsymbol{b}^{k-1}) - \mathsf{F}(\overline{\boldsymbol{b}})\|_2 + \|\widehat{\mathsf{F}}(\boldsymbol{b}^{k-1}) - \mathsf{F}(\boldsymbol{b}^{k-1})\|_2 \\
&\leq \eta\|\boldsymbol{b}^{k-1} - \overline{\boldsymbol{b}}\|_2 + \|\widehat{\mathsf{F}}(\boldsymbol{b}^{k-1}) - \mathsf{F}(\boldsymbol{b}^{k-1})\|_2,
\end{aligned}$$

where we first used the triangular inequality and then the fact that $\mathsf{F}$ is a contraction. By taking the conditional expectation on both sides, we obtain

$$\mathbb{E}\left[\|\boldsymbol{b}^k - \overline{\boldsymbol{b}}\|_2 \mid \boldsymbol{x}^K, \boldsymbol{b}^{k-1}\right] \leq \eta\|\boldsymbol{b}^{k-1} - \overline{\boldsymbol{b}}\|_2 + \mathbb{E}\left[\|\widehat{\mathsf{F}}(\boldsymbol{b}^{k-1}) - \mathsf{F}(\boldsymbol{b}^{k-1})\|_2 \mid \boldsymbol{x}^K, \boldsymbol{b}^{k-1}\right]. \quad (3)$$

We can bound the second term in (3) as follows

$$\mathbb{E}\left[\|\widehat{\mathsf{F}}(\boldsymbol{b}^{k-1}) - \mathsf{F}(\boldsymbol{b}^{k-1})\|_2 \mid \boldsymbol{x}^K, \boldsymbol{b}^{k-1}\right] \leq \gamma\mathbb{E}\left[\|\mathsf{H}g(\overline{\boldsymbol{x}}) - \mathsf{H}\widehat{g}(\boldsymbol{x}^K)\|_2 \mid \boldsymbol{x}^K, \boldsymbol{b}^{k-1}\right]\|\boldsymbol{b}^{k-1}\|_2$$

$$+ \gamma\tau\|\nabla_{\boldsymbol{x}}\mathsf{R}(\overline{\boldsymbol{x}}) - \nabla_{\boldsymbol{x}}\mathsf{R}(\boldsymbol{x}^K)\|_2\|\boldsymbol{b}^{k-1}\|_2 + \|\boldsymbol{x}^K - \overline{\boldsymbol{x}}\|_2$$

$$\leq \gamma\lambda\|\boldsymbol{x}^K - \overline{\boldsymbol{x}}\|_2\|\boldsymbol{b}^{k-1}\|_2 + \frac{\gamma\nu}{\sqrt{w}}\|\boldsymbol{b}^{k-1}\|_2 + \gamma\tau\alpha\|\boldsymbol{x}^K - \overline{\boldsymbol{x}}\|_2\|\boldsymbol{b}^{k-1}\|_2 + \|\boldsymbol{x}^K - \overline{\boldsymbol{x}}\|_2$$

$$\leq A_1\|\boldsymbol{x}^K - \overline{\boldsymbol{x}}\|_2 + \frac{A_2}{\sqrt{w}},$$

with $A_1 := (2\gamma\lambda + 2\gamma\tau\alpha R + 1)$ and $A_2 = 2\nu R$, where in the second inequality we used Lemma 2 and the $\alpha$-Lipschitz continuity of $\nabla_{\boldsymbol{x}}\mathsf{R}$ and in the third $\|\boldsymbol{b}^{k-1}\|_2 \leq 2R$. Since $\boldsymbol{b}^0 = \boldsymbol{0}$, Assumption 3 implies that $\|\overline{\boldsymbol{b}}\|_2 \leq R$, which leads to $\|\boldsymbol{b}^{k-1}\|_2 \leq 2R$ for all $k \geq 1$.

By including the last bound into (3), we obtain

$$\mathbb{E}\left[\|\boldsymbol{b}^k - \overline{\boldsymbol{b}}\|_2 \mid \boldsymbol{x}^K, \boldsymbol{b}^{k-1}\right] \leq \eta\|\boldsymbol{b}^{k-1} - \overline{\boldsymbol{b}}\|_2 + A_1\|\boldsymbol{x}^K - \overline{\boldsymbol{x}}\|_2 + A_2/\sqrt{w}.$$

By taking the total expectation and using Proposition 1, we get

$$\mathbb{E}\left[\|\boldsymbol{b}^k - \overline{\boldsymbol{b}}\|_2\right] \leq \eta\mathbb{E}\left[\|\boldsymbol{b}^{k-1} - \overline{\boldsymbol{b}}\|_2\right] + A_1 R\eta^K + \frac{A_1\gamma\nu}{(1-\nu)\sqrt{w}} + \frac{A_2}{\sqrt{w}}.$$

By iterating this bound and noting that $k \leq K$, we get the final result

$$\mathbb{E}\left[\|\boldsymbol{b}^k - \overline{\boldsymbol{b}}\|_2\right] \leq \eta^k B_1 + \frac{B_2}{\sqrt{w}},$$

where $B_1 := R + A_1 R/(1-\eta)$ and $B_2 := (((A_1\gamma\nu)/(1-\eta)) + A_2)/(1-\nu)$.

$\square$

## B.1 Technical Lemma for the Proof of Proposition 2

The following technical result is used in the proof of Proposition 2. It bounds the variance of the Hessian of the data-fidelity term $g$.

**Lemma 2.** *Suppose that Assumptions 1 and 4 in the main paper are true. Then, for any $\boldsymbol{z}, \boldsymbol{y} \in \mathbb{R}^n$*

$$\mathbb{E}\left[\|\mathsf{H}g(\boldsymbol{z}) - \mathsf{H}\widehat{g}(\boldsymbol{y})\|_2\right] \leq \lambda\|\boldsymbol{z} - \boldsymbol{y}\|_2 + \frac{\nu}{\sqrt{w}},$$

*where the expectation is taken over the indices $\{i_1, \ldots, i_w\}$ used for $\widehat{g}$.*

*Proof.* The proof directly follows the $\lambda$-Lipschitz continuity assumption of $\mathsf{H}g$ is Assumption 1 and boundedness of the variance in Assumption 4

$$\mathbb{E}\left[\|\mathsf{H}g(\boldsymbol{z}) - \mathsf{H}\widehat{g}(\boldsymbol{y})\|_2\right] \leq \mathbb{E}\left[\|\mathsf{H}g(\boldsymbol{z}) - \mathsf{H}g(\boldsymbol{y})\|_2\right] + \mathbb{E}\left[\|\mathsf{H}g(\boldsymbol{y}) - \mathsf{H}\widehat{g}(\boldsymbol{y})\|_2\right]$$

$$\leq \lambda\|\boldsymbol{z} - \boldsymbol{y}\|_2 + \frac{\nu}{\sqrt{w}},$$

where we used the Jensen's inequality to get the second term.

$\square$

## C   Proof of Main Theorem

Our main theoretical result is a novel analysis on the ability of ODER to approximate the stationary points of the *training loss $\ell(\boldsymbol{\theta})$*. We show that ODER can approximate (in expectation) the stationary points up to an error term that can be controlled by the minibatch size $w$ and the learning rate $\beta$.

**Main Theorem.** *Train ODER using SGD for $T \geq 1$ iterations under Assumptions 1-6 using the step-size parameters $0 < \beta \leq 1/L$ and the minibatch size $w \geq 1$. Select a large enough number of forward and backward pass iterations $K \geq 1$ to satisfy $0 < \eta^K \leq 1/\sqrt{w}$. Then, we have that*

$$\frac{1}{T} \sum_{t=0}^{T-1} \mathbb{E}\left[\|\nabla\ell(\boldsymbol{\theta}^t)\|_2^2\right] \leq \frac{2(\ell(\boldsymbol{\theta}^0) - \ell(\boldsymbol{\theta}^*))}{\beta T} + \frac{C_1}{\sqrt{w}} + \beta C_2.$$

*where $C_1 > 0$ and $C_2 > 0$ are constants independent of $T$ and $w$.*

*Proof.* Consider the RED (DEQ) loss $\ell$ and its ODER approximation $\hat{\ell}$

$$\ell(\boldsymbol{\theta}) = \frac{1}{p} \sum_{j=1}^{p} \ell_j(\boldsymbol{\theta}) \quad \text{and} \quad \hat{\ell}(\boldsymbol{\theta}) = \frac{1}{p} \sum_{j=1}^{p} \hat{\ell}_j(\boldsymbol{\theta}), \tag{4}$$

where each $\ell_j$ and $\hat{\ell}_j$ have the forms

$$\ell_j(\boldsymbol{\theta}) = \frac{1}{2}\|\overline{\boldsymbol{x}}_j(\boldsymbol{\theta}) - \boldsymbol{x}_j^*\|_2^2 \quad \text{and} \quad \hat{\ell}_j(\boldsymbol{\theta}) = \frac{1}{2}\|\boldsymbol{x}_j^K(\boldsymbol{\theta}) - \boldsymbol{x}_j^*\|_2^2.$$

Vector $\boldsymbol{x}_j^K$ denotes the final iterate of the online forward pass obtained after $K \geq 1$ iterations for the training sample $j \in \{1, \cdots, p\}$.

From Assumption 5, we obtain the traditional Lipschitz continuity bound on the gradient

$$\|\nabla\ell(\boldsymbol{\theta}_1) - \nabla\ell(\boldsymbol{\theta}_2)\|_2 \leq L\|\boldsymbol{\theta}_1 - \boldsymbol{\theta}_2\|_2,$$

which directly leads to traditional quadratic upper bound (see Lemma 1.2.3 in [8]).

$$\ell(\boldsymbol{\theta}_1) \leq \ell(\boldsymbol{\theta}_2) + \nabla\ell(\boldsymbol{\theta}_2)^{\mathsf{T}}(\boldsymbol{\theta}_1 - \boldsymbol{\theta}_2) + \frac{L}{2}\|\boldsymbol{\theta}_1 - \boldsymbol{\theta}_2\|_2^2. \tag{5}$$

Lemma 3 in this supplement establishes the following useful bound for our analysis

$$\mathbb{E}\left[\|\nabla\hat{\ell}(\boldsymbol{\theta}^t) - \nabla\ell(\boldsymbol{\theta}^t)\|_2^2\right] \leq \frac{C}{\sqrt{w}},$$

for some constant $C > 0$ (see its full expression in Lemma 3). This directly implies that

$$\mathbb{E}\left[-\nabla\ell(\boldsymbol{\theta}^t)^{\mathsf{T}}\nabla\hat{\ell}(\boldsymbol{\theta}^t) + \frac{1}{2}\|\nabla\hat{\ell}(\boldsymbol{\theta}^t)\|_2^2\right] \leq -\frac{1}{2}\mathbb{E}[\|\nabla\ell(\boldsymbol{\theta}^t)\|_2^2] + \frac{C}{2\sqrt{w}}. \tag{6}$$

Consider a single iteration of SGD for optimizing ODER

$$\boldsymbol{\theta}^{t+1} = \boldsymbol{\theta}^t - \beta\nabla\hat{\ell}_{j_t}(\boldsymbol{\theta}^t). \tag{7}$$

From the quadratic upper bound (5), we get

$$\ell(\boldsymbol{\theta}^{t+1}) - \ell(\boldsymbol{\theta}^t) \leq \nabla\ell(\boldsymbol{\theta}^t)^{\mathsf{T}}(\boldsymbol{\theta}^{t+1} - \boldsymbol{\theta}^t) + \frac{L}{2}\|\boldsymbol{\theta}^{t+1} - \boldsymbol{\theta}^t\|_2^2$$

$$= -\beta\nabla\ell(\boldsymbol{\theta}^t)^{\mathsf{T}}\nabla\hat{\ell}_{j_t}(\boldsymbol{\theta}^t) + \frac{\beta^2 L}{2}\|\nabla\hat{\ell}_{j_t}(\boldsymbol{\theta}^t)\|_2^2.$$

For notation convenience, we use $\mathbb{E}[\cdot \,|\, \backslash j_t]$ to denote the expectation only with respect to the training index $j_t \in \{1, \ldots, p\}$, where we condition on $\boldsymbol{\theta}^t$ and the random indices within forward and backward passes at this iteration. By taking $\mathbb{E}[\cdot \,|\, \backslash j_t]$ on both sides of the quadratic upper bound (5)

$$\mathbb{E}\left[\ell(\boldsymbol{\theta}^{t+1})| \backslash j_t\right] - \ell(\boldsymbol{\theta}^t) \leq -\beta\nabla\ell(\boldsymbol{\theta}^t)^{\mathsf{T}}\mathbb{E}\left[\nabla\hat{\ell}_{j_t}(\boldsymbol{\theta}^t)| \backslash j_t\right] + \frac{\beta^2 L}{2}\mathbb{E}\left[\|\nabla\hat{\ell}_{j_t}(\boldsymbol{\theta}^t)\|_2^2| \backslash j_t\right]$$

$$= -\beta\nabla\ell(\boldsymbol{\theta}^t)^{\mathsf{T}}\nabla\hat{\ell}(\boldsymbol{\theta}^t) + \frac{\beta^2 L}{2}\mathbb{E}\left[\|\nabla\hat{\ell}_{j_t}(\boldsymbol{\theta}^t)\|_2^2| \backslash j_t\right], \tag{8}$$

where we use (4) and the fact that $j_t$ is distributed uniformly at random in $\{1, \ldots, p\}$.

We now estimate the last term in (8)

$$
\begin{aligned}
&\mathbb{E}\left[\|\nabla\hat{\ell}_{j_t}(\boldsymbol{\theta}^t)\|_2^2| \setminus j_t\right]\\
&= \mathbb{E}\left[\|\nabla\hat{\ell}_{j_t}(\boldsymbol{\theta}^t) - \nabla\hat{\ell}(\boldsymbol{\theta}^t) + \nabla\hat{\ell}(\boldsymbol{\theta}^t)\|_2^2| \setminus j_t\right]\\
&= \mathbb{E}\left[\|\nabla\hat{\ell}_{j_t}(\boldsymbol{\theta}^t) - \nabla\hat{\ell}(\boldsymbol{\theta}^t)\|_2^2 + 2(\nabla\hat{\ell}_{j_t}(\boldsymbol{\theta}^t) - \nabla\hat{\ell}(\boldsymbol{\theta}^t))^\mathsf{T}\nabla\hat{\ell}(\boldsymbol{\theta}^t) + \|\nabla\hat{\ell}(\boldsymbol{\theta}^t)\|_2^2| \setminus j_t\right]\\
&= \mathbb{E}\left[\|\nabla\hat{\ell}_{j_t}(\boldsymbol{\theta}^t) - \nabla\hat{\ell}(\boldsymbol{\theta}^t)\|_2^2| \setminus j_t\right] + \mathbb{E}\left[\|\nabla\hat{\ell}(\boldsymbol{\theta}^t)\|_2^2| \setminus j_t\right],
\end{aligned}
\tag{9}
$$

where in the third equality we use the fact that $\mathbb{E}\left[(\nabla\hat{\ell}_{j_t}(\boldsymbol{\theta}^t) - \nabla\hat{\ell}(\boldsymbol{\theta}^t))^\mathsf{T}\nabla\hat{\ell}(\boldsymbol{\theta}^t)| \setminus j_t\right] = 0.$

By replacing the last term in (8) with (9) and taking the full expectation on both sides of (8) in terms of all random variables and invoking the bound (6), we obtain

$$
\begin{aligned}
&\mathbb{E}[\ell(\boldsymbol{\theta}^{t+1})] - \mathbb{E}\left[\ell(\boldsymbol{\theta}^t)\right]\\
&\leq \mathbb{E}\left[-\beta\nabla\ell(\boldsymbol{\theta}^t)^\mathsf{T}\nabla\hat{\ell}(\boldsymbol{\theta}^t) + \frac{\beta^2 L}{2}\|\nabla\hat{\ell}(\boldsymbol{\theta}^t)\|\right] + \frac{\beta^2 L}{2}\mathbb{E}\left[\|\nabla\hat{\ell}_{j_t}(\boldsymbol{\theta}^t) - \nabla\hat{\ell}(\boldsymbol{\theta}^t)\|_2^2\right]\\
&\leq -\frac{\beta}{2}\mathbb{E}[\|\nabla\ell(\boldsymbol{\theta}^t)\|_2^2] + \frac{\beta C}{2\sqrt{w}} + \frac{\beta^2 L}{2}(4\sigma^2 + \frac{6C}{\sqrt{w}}),
\end{aligned}
$$

where we used the fact that $0 < \beta \leq 1/L$, and in the last inequality we used the bound from Lemma 4 in this document. By rearranging the terms, and summing this bound over $0 \leq t \leq T - 1$, we obtain

$$
\frac{1}{T}\sum_{t=0}^{T-1}\mathbb{E}\left[\|\nabla\ell(\boldsymbol{\theta}^t)\|_2^2\right] \leq \frac{2(\ell(\boldsymbol{\theta}^0) - \mathbb{E}[\ell(\boldsymbol{\theta}^T)])}{\beta T} + \frac{C_1}{\sqrt{w}} + \beta C_2
$$

$$
\leq \frac{2(\ell(\boldsymbol{\theta}^0) - \ell(\boldsymbol{\theta}^*))}{\beta T} + \frac{C_1}{\sqrt{w}} + \beta C_2,
$$

where $C_1 := 7\gamma^2\tau^2\alpha^2(R + R^2)(B_1 + B_2 + R^2 + \gamma\nu R/(1 - \eta))$ and $C_2 := 4L\sigma^2$ are constants independent of $t$ and $w$. In the last inequality we used the fact that $\ell(\boldsymbol{\theta}^*) \leq \ell(\boldsymbol{\theta})$ for all $\boldsymbol{\theta}$, where $\boldsymbol{\theta}^*$ denotes a global minimizer of $\ell$. $\qquad\square$

## C.1 Technical Lemmas for the Proof of Main Theorem

The following lemma are useful for relating $\nabla\hat{\ell}$ and $\nabla\ell$ in expectation up to an error term. Both are used in the proof of Main Theorem.

**Lemma 3.** *Given the loss function $\ell$ of RED (DEQ) and $\hat{\ell}$ of ODER, by selecting a large enough number of forward and backward iterations $K \geq 1$ to satisfy $0 < \eta^K \leq 1/\sqrt{w}$, we have*

$$
\mathbb{E}\left[\|\nabla\hat{\ell}(\boldsymbol{\theta}) - \nabla\ell(\boldsymbol{\theta})\|_2^2\right] \leq \frac{C}{\sqrt{w}},
$$

*where $C := \gamma^2\tau^2\alpha^2(R + R^2)(B_1 + B_2 + R^2 + \gamma\nu R/(1 - \eta))$ is a constant.*

*Proof.* By using the DEQ training, we have the following bound

$$
\begin{aligned}
\|\nabla\hat{\ell}_j(\boldsymbol{\theta}) - \nabla\ell_j(\boldsymbol{\theta})\|_2 &= \left\|\left[\nabla_{\boldsymbol{\theta}}\widehat{\mathsf{T}}_{\boldsymbol{\theta}}(\boldsymbol{x}_j^K)\right]^\mathsf{T}\boldsymbol{b}_j^K - [\nabla_{\boldsymbol{\theta}}\mathsf{T}_{\boldsymbol{\theta}}(\overline{\boldsymbol{x}}_j)]^\mathsf{T}\overline{\boldsymbol{b}}_j\right\|_2\\
&= \left\|\left[\nabla_{\boldsymbol{\theta}}\widehat{\mathsf{T}}_{\boldsymbol{\theta}}(\boldsymbol{x}_j^K)\right]^\mathsf{T}(\boldsymbol{b}_j^K - \overline{\boldsymbol{b}}_j) + \left[\nabla_{\boldsymbol{\theta}}\widehat{\mathsf{T}}_{\boldsymbol{\theta}}(\boldsymbol{x}_j^K) - \nabla_{\boldsymbol{\theta}}\mathsf{T}_{\boldsymbol{\theta}}(\overline{\boldsymbol{x}}_j)\right]^\mathsf{T}\overline{\boldsymbol{b}}_j\right\|_2\\
&\leq \left\|\nabla_{\boldsymbol{\theta}}\widehat{\mathsf{T}}_{\boldsymbol{\theta}}(\boldsymbol{x}_j^K)\right\|_2\|\boldsymbol{b}_j^K - \overline{\boldsymbol{b}}_j\|_2 + \left\|\nabla_{\boldsymbol{\theta}}\widehat{\mathsf{T}}_{\boldsymbol{\theta}}(\boldsymbol{x}_j^K) - \nabla_{\boldsymbol{\theta}}\mathsf{T}_{\boldsymbol{\theta}}(\overline{\boldsymbol{x}}_j)\right\|_2\|\overline{\boldsymbol{b}}_j\|_2\\
&\leq \gamma\tau\alpha(\|\boldsymbol{b}_j^K - \overline{\boldsymbol{b}}_j\|_2 + \|\boldsymbol{x}_j^K - \overline{\boldsymbol{x}}_j\|_2\|\overline{\boldsymbol{b}}_j\|_2), \quad \forall j \in \{1, \cdots, p\},
\end{aligned}
\tag{10}
$$

where in the first inequality, we used Cauchy-Schwarz inequality and the fact that $\mathsf{D}_{\boldsymbol{\theta}}$ and $\nabla_{\boldsymbol{\theta}}\mathsf{D}_{\boldsymbol{\theta}}(\boldsymbol{x})$ are $\alpha$-Lipschitz continuous with respect to $\boldsymbol{x}$ and $\boldsymbol{\theta}$ based on Assumption 2. By applying Assumption 3 which states that $\|\boldsymbol{b}^k - \overline{\boldsymbol{b}}\|_2 < R$ and $\|\boldsymbol{x}^k - \overline{\boldsymbol{x}}\|_2 < R$ for every $\boldsymbol{x}, \boldsymbol{b} \in \mathbb{R}^n$, we have

$$
\|\nabla\hat{\ell}_j(\boldsymbol{\theta}) - \nabla\ell_j(\boldsymbol{\theta})\|_2 \leq \gamma\tau\alpha(R + R^2), \quad \forall j \in \{1, \cdots, p\}
\tag{11}
$$

On the other hand, by taking the expectation with respect to the stochastic approximation variables in forward and backward propagation in (10), invoking the bounds obtained from Proposition 1 and Proposition 2 and using the fact that $0 < \eta^K \leq 1/\sqrt{w}$, we have

$$\mathbb{E}\left[\|\nabla\hat{\ell}_j(\boldsymbol{\theta}) - \nabla\ell_j(\boldsymbol{\theta})\|_2\right] \leq \gamma\tau\alpha\left(\frac{B_1 + B_2 + R^2}{\sqrt{w}} + \frac{\gamma\nu R}{(1-\eta)\sqrt{w}}\right)$$

where $B_1 > 0$ and $B_2 > 0$ are constants obtained in Proposition 2. Now consider a random variable $X$ which has a probability density function given by $f(x)$ on the real number line such that $P(0 \leq X \leq c) = 1$, then we have

$$\mathbb{E}[X^2] = \int_0^c x^2 f(x)dx \leq \int_0^c cxf(x)dx = c\mathbb{E}[X].$$

As a consequence, given the bound (11) for the random variables $\|\nabla\hat{\ell}_j(\boldsymbol{\theta}) - \nabla\ell_j(\boldsymbol{\theta})\|_2$ and using the above fact, we have the following useful bound for our proof

$$\mathbb{E}\left[\|\nabla\hat{\ell}_j(\boldsymbol{\theta}) - \nabla\ell_j(\boldsymbol{\theta})\|_2^2\right] \leq \frac{C}{\sqrt{w}}, \tag{12}$$

where $C := \gamma^2\tau^2\alpha^2(R + R^2)(B_1 + B_2 + R^2 + \gamma\nu R/(1-\eta))$ is a constant.

Note also the fact that for $\boldsymbol{a}_1, \cdots, \boldsymbol{a}_p \in \mathbb{R}^n$, we have

$$\left\|\sum_{j=1}^p \boldsymbol{a}_j\right\|_2^2 \leq p\sum_{j=1}^p \|\boldsymbol{a}_j\|_2^2.$$

As a result, applying the bound (12), we have

$$\begin{aligned}
\mathbb{E}\left[\|\nabla\hat{\ell}(\boldsymbol{\theta}) - \nabla\ell(\boldsymbol{\theta})\|_2^2\right] &= \frac{1}{p^2}\mathbb{E}\left[\left\|\sum_{j=1}^p(\nabla\hat{\ell}_j(\boldsymbol{\theta}) - \nabla\ell_j(\boldsymbol{\theta}))\right\|_2^2\right] \\
&\leq \frac{1}{p}\sum_{j=1}^p \mathbb{E}\left[\left\|\nabla\hat{\ell}_j(\boldsymbol{\theta}) - \nabla\ell_j(\boldsymbol{\theta})\right\|_2^2\right] \\
&\leq \frac{C}{\sqrt{w}}.
\end{aligned}$$

This finishes the proof. $\qquad\square$

**Lemma 4.** *Given the loss function $\ell$ and $\hat{\ell}$, by taking the conditional expectation with respect to the training index $j_t$ (via conditioning on all other variables), we have*

$$\mathbb{E}\left[\|\nabla\hat{\ell}_{j_t}(\boldsymbol{\theta}^t) - \nabla\hat{\ell}(\boldsymbol{\theta}^t)\|_2^2\right] \leq 4\sigma^2 + \frac{6C}{\sqrt{w}},$$

*where $C > 0$ obtained in Lemma 3 is a constant.*

*Proof.* By taking expectation with respect to training index $j_t \in \{1\cdots, N\}$, we obtain the following useful bound for our proof

$$\begin{aligned}
&\mathbb{E}\left[\|\nabla\ell_{j_t}(\boldsymbol{\theta}^t) - \nabla\hat{\ell}(\boldsymbol{\theta}^t)\|_2^2| \setminus j_t\right] \\
&= \mathbb{E}\left[\|\nabla\ell_{j_t}(\boldsymbol{\theta}^t) - \nabla\ell(\boldsymbol{\theta}^t) + \nabla\ell(\boldsymbol{\theta}^t) - \nabla\hat{\ell}(\boldsymbol{\theta}^t)\|_2^2| \setminus j_t\right] \\
&= \mathbb{E}\left[\|\nabla\ell_{j_t}(\boldsymbol{\theta}^t) - \nabla\ell(\boldsymbol{\theta}^t)\|_2^2 + \|\nabla\ell(\boldsymbol{\theta}^t) - \nabla\hat{\ell}(\boldsymbol{\theta}^t)\|_2^2 \right. \\
&\qquad \left. + 2(\nabla\ell_{j_t}(\boldsymbol{\theta}^t) - \nabla\ell(\boldsymbol{\theta}^t))^{\mathsf{T}}(\nabla\ell(\boldsymbol{\theta}^t) - \nabla\hat{\ell}(\boldsymbol{\theta}^t))| \setminus j_t\right] \\
&\leq 2\left(\mathbb{E}\left[\|\nabla\ell_{j_t}(\boldsymbol{\theta}^t) - \nabla\ell(\boldsymbol{\theta}^t)\|_2^2 + \|\nabla\ell(\boldsymbol{\theta}^t) - \nabla\hat{\ell}(\boldsymbol{\theta}^t)\|_2^2| \setminus j_t\right]\right) \\
&= 2\mathbb{E}\left[\|\nabla\ell_{j_t}(\boldsymbol{\theta}^t) - \nabla\ell(\boldsymbol{\theta}^t)\|_2^2| \setminus j_t\right] + 2\|\nabla\ell(\boldsymbol{\theta}^t) - \nabla\hat{\ell}(\boldsymbol{\theta}^t)\|_2^2,
\end{aligned} \tag{13}$$

where we used Young's inequality that states for any $\boldsymbol{a}_1, \boldsymbol{a}_2 \in \mathbb{R}^n$, we have

$$2\boldsymbol{a}_1^{\mathsf{T}}\boldsymbol{a}_2 \leq \|\boldsymbol{a}_1\|_2^2 + \|\boldsymbol{a}_2\|_2^2 \quad \Rightarrow \quad \|\boldsymbol{a}_1 + \boldsymbol{a}_2\|_2^2 \leq 2(\|\boldsymbol{a}_1\|_2^2 + \|\boldsymbol{a}_2\|_2^2).$$

By taking the full expectation of the inequality (13) above and applying Lemma 3 and the bounded variance in Assumption 6, we have

$$
\begin{aligned}
&\mathbb{E}\left[\|\nabla\ell_{j_t}(\boldsymbol{\theta}^t) - \nabla\hat{\ell}(\boldsymbol{\theta}^t)\|_2^2\right] \\
&\leq 2\mathbb{E}\left[\mathbb{E}\left[\|\nabla\ell_{j_t}(\boldsymbol{\theta}^t) - \nabla\ell(\boldsymbol{\theta}^t)\|_2^2 | \setminus j_t\right]\right] + 2\mathbb{E}\left[\|\nabla\hat{\ell}(\boldsymbol{\theta}^t) - \nabla\ell(\boldsymbol{\theta}^t)\|_2^2\right] \\
&\leq 2\sigma^2 + \frac{2C}{\sqrt{w}}
\end{aligned}
\tag{14}
$$

Similarly, by taking full expectation and using Lemma 3 and the bound (14), we write that

$$
\begin{aligned}
&\mathbb{E}\left[\|\nabla\hat{\ell}_{j_t}(\boldsymbol{\theta}^t) - \nabla\hat{\ell}(\boldsymbol{\theta}^t)\|_2^2\right] \\
&= \mathbb{E}\left\|\nabla\hat{\ell}_{j_t}(\boldsymbol{\theta}^t) - \nabla\ell_{j_t}(\boldsymbol{\theta}^t) + \nabla\ell_{j_t}(\boldsymbol{\theta}^t) - \nabla\hat{\ell}(\boldsymbol{\theta}^t)\|_2^2\right] \\
&\leq 2\mathbb{E}\left[\|\nabla\ell_{j_t}(\boldsymbol{\theta}^t) - \nabla\hat{\ell}(\boldsymbol{\theta}^t)\|_2^2\right] + 2\mathbb{E}\left[\|\nabla\hat{\ell}_{j_t}(\boldsymbol{\theta}^t) - \nabla\ell_{j_t}(\boldsymbol{\theta}^t)\|_2^2\right] \\
&\leq 4\sigma^2 + \frac{6C}{\sqrt{w}}.
\end{aligned}
$$

$\square$

## D  Additional Technical Details and Numerical Results

In this section, we present technical details that were omitted from the main paper for space. We used the following *signal-to-noise ratio (SNR)* [4, 9] in dB for quantitively comparing different algorithms

$$\text{SNR}(\widehat{\boldsymbol{x}}, \boldsymbol{x}) = \max_{a, b \in \mathbb{R}} \left\{ 20\log_{10}\left(\frac{\|\boldsymbol{x}\|_2}{\|\boldsymbol{x} - a\widehat{\boldsymbol{x}} + b\|_2}\right) \right\}, \tag{15}$$

where $\widehat{\boldsymbol{x}}$ and $\boldsymbol{x}$ represents the noisy vector and ground truth respectively, while the purpose of $a$ and $b$ is to adjust for contrast and offset. We also used the *structural similarity index measure (SSIM)* [10] as an alternative metric. All the experiments in this work were performed on a machine equipped with an Intel Xeon Gold 6130 Processor and eight NVIDIA GeForce RTX 2080 Ti GPUs.

As stated in [7, 11] and other DEQ work, using acceleration can reduce computational costs during both training and inference time and lead to improvement of empirical performance at inference. Here, we focus on the final image reconstruction performance for denoising based step-descent RED (SD-RED) by using two different fixed-point acceleration methods, namely Anderson acceleration and Nesterov acceleration. The detailed instructions of using Anderson acceleration is publicly available with tutorials [1]. The Nesterov acceleration for RED (DEQ) and RED (Denoising) can be summarized as

$$
\begin{aligned}
\boldsymbol{x}^k &= \mathsf{T}_{\boldsymbol{\theta}}(\boldsymbol{s}^k) \\
c_k &= (q_{k-1} - 1)/q_k \\
\boldsymbol{s}^k &= \boldsymbol{x}^k + c_k(\boldsymbol{x}^k - \boldsymbol{x}^{k-1}) \,,
\end{aligned}
$$

where the value of $q_k = 1/2(1 + \sqrt{1 + 4q_{k-1}^2})$ is adapted for better PSNR performance. The average SNR (dB) values for RED (Anderson) and RED (Nesterov) using different CNN denoisers on the MRI images are presented in Table 1. We empirically observe that the RED with Nesterov acceleration led to better reconstructions in terms of SNR. For the forward pass iterations, we equip ODER, RED (Denoising), RED (Unfold) and RED (DEQ) with Nesterov acceleration for all experiments used in this work. We utilize Anderson acceleration for the backward pass for both ODER and RED (DEQ). We limit the number of backward pass iterations to 50 for efficiency considerations for all three imaging applications. The number of forward passes is presented in each imaging modality

---

[1]Anderson acceleration for DEQ was introduced at `http://implicit-layers-tutorial.org/`.

Table 1: Average SNR (dB) for different pre-trained CNNs on MRI test images. Note that the "AWGN denoising" performance is for noise level $\sigma = 5$ and the "Time (ms)" presents the runtime of evaluating $\mathsf{R}_{\boldsymbol{\theta}}(\boldsymbol{x})/\nabla_{\boldsymbol{x}}\mathsf{R}_{\boldsymbol{\theta}}(\boldsymbol{x})$ on images of size $320 \times 320$.

| Model ⁞ SNR(dB) | DnCNN | Tiny U-Net | U-Net |
|---|---|---|---|
| **AWGN denoising** | 30.30 | 30.36 | 30.41 |
| **RED (Nesterov)** | 26.37 | 26.35 | 26.42 |
| **RED (Anderson)** | 25.44 | 25.46 | 25.51 |
| **Time (ms)** | 12.22 / 31.74 | 1.65 / 11.48 | 1.88 / 32.84 |

sub-section, respectively. Followed by [7, 12], we additionally set the convergence criterion (relative norm difference between iterations) as

$$\frac{\|\boldsymbol{x}^{k+1} - \boldsymbol{x}^{k}\|_2}{\|\boldsymbol{x}^{k}\|_2} < \epsilon,$$

where $\epsilon > 0$. In forward passes, We set $\epsilon = 10^{-3}$ for ODER and RED (DEQ), while we set stopping criterion of backward passes to $\epsilon = 10^{-2}$ for ODER and RED (DEQ).

We additionally tested three network architectures including DnCNN [13], U-Net [14] and tiny U-Net [15]. The DnCNN network has seventeen layers, including 15 hidden layers, an input layer, and an output layer. The tiny U-Net is a simplified variant of the normal U-Net with less trainable parameters. In specific, the CNN consists of four scales, each with a skip connection between downsampling and upsampling. These connections increase the effective receptive field of the CNN. The number of channels in each layer are $\{32, 64, 128, 256\}$. We make two additional modifications to the tiny U-Net. First, we drop out the second group normalization (GN) [16] at each composite convolutional layers. Second, we add spectral normalization to each layers for more stable training and better Lipschitz constrain of the neural network. It is worth to note that spectral normalization is a widely used method for Lipschitz constrained neural network, and it is *not* our aim to claim any algorithmic novelty with respect to it. In Table 1, we present the denoising performance on AWGN removal with noise level $\sigma = 5$ and the run time of calculating $\mathsf{R}_{\boldsymbol{\theta}}(\boldsymbol{x})/\nabla_{\boldsymbol{x}}\mathsf{R}_{\boldsymbol{\theta}}(\boldsymbol{x})$ with respect to a $320 \times 320$ image. Overall, the traditional U-Net architecture achieves the best denoising and reconstruction performance, but requires more time per iterate than tiny U-Net. As a result, we implement traditional U-Net denoiser for RED (Denoising), and we equip ODER, RED (DEQ) and RED (Unfold) for the same tiny U-Net architecture in order to decrease per-iteration computation costs during training.

In Fig. 1, we report the empirical evaluation of the Lipschitz constant $\eta$ of $\mathsf{T}$ used in our simulations and stated in Lemma 1 on the testing images from all three inverse problems in the main paper. We plot the histograms of values $\eta = \|\mathsf{T}(\boldsymbol{x}^{k-1}) - \mathsf{T}(\overline{\boldsymbol{x}})\|_2/\|\boldsymbol{x}^{k-1} - \overline{\boldsymbol{x}}\|_2$, and the maximum value of each histogram is indicated by a vertical bar with the frequency of $\eta > 1$, providing an empirical upper bound on the values of $\eta$. Note that despite the numerical limitations of current spectral normalization techniques, they still provide a useful tool to ensure stable convergence.

### D.1 Additional Details and Validations for IDT

We follow the experimental setup in [4, 6, 17] to generate the measurement matrix and simulated images for IDT [2]. In specific, the simulated images are assumed to be on the focal plane $z = 0\mu\mathrm{m}$ with LEDs located at $z_{\mathrm{LED}} = -70\mathrm{mm}$. The wavelength of the illumination was set to $\lambda = 630\mathrm{nm}$ and the background medium index was assumed to be water with $\epsilon_b = 1.33$. We generated $b = 500$ intensity measurements with $40\times$ microscope objectives (MO) and 0.65 numerical aperture (NA). Followed by [6], we assume real permittivity function, and our implementation stores each $\boldsymbol{A}_i$ as two separate arrays for phase and absorption. In addition, each matrix is stored in the Fourier space to reduce the computational complexity of evaluating convolutions. This result in the storage of complex valued arrays for each, consisting of pairs of single precision floats for every element when training ODER/RED (DEQ). Thus, the shape of each measurements and measurement operators in

---

[2] The code is publicly available at `https://github.com/bu-cisl/High-Throughput-IDT`

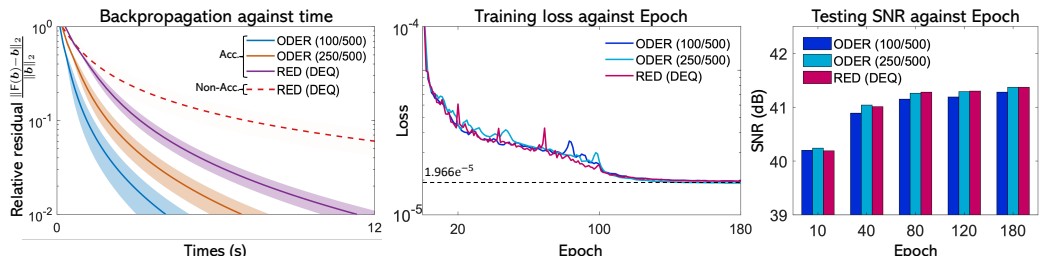

Figure 2: *Numerical Illustration of ODER on IDT for two minibatch sizes $w \in \{100, 250\}$. The result for RED (DEQ) with $b = 500$ is also provided for reference. The left figure shows how ODER improves the efficiency of the backward pass of RED (DEQ) by reducing the per-iteration complexity of the measurement matrix. The middle figure plots the loss against the epoch number evaluated on the training set. The right figure plots the SNR (dB) achieved at different epochs for ODER evaluated over the testing set. This figure highlights that by using minibatches $1 \leq w \leq b$, ODER improves per-iteration complexity and matches the same final imaging quality achieved by RED (DEQ).*

Table 2: Average SSIM values for IDT image recovery on testing images from [18].

| Method | Input SNR (dB) | | |
|---|---|---|---|
| | 15 | 20 | 25 |
| TV | 0.9810 | 0.9829 | 0.9835 |
| U-Net | 0.9811 | 0.9831 | 0.9836 |
| ISTA-Net+ | 0.9809 | 0.9833 | 0.9841 |
| SGD-Net (100) | 0.9832 | 0.9859 | 0.9866 |
| RED (Denoising) | 0.9831 | 0.9852 | 0.9866 |
| ODER (100) | 0.9834 | 0.9875 | 0.9889 |
| ODER (250) | **0.9846** | 0.9878 | **0.9891** |
| RED (DEQ) | 0.9845 | **0.9879** | 0.9890 |

full batch RED (DEQ) for reconstructing one slice is $1 \times 416 \times 416 \times 500 \times 2$. A detailed discussion on the IDT forward model is available in [4, 17].

We train ODER and RED (DEQ) with the initialization $\boldsymbol{x}_0 = \boldsymbol{A}^\mathsf{H}\boldsymbol{y}$, where $\boldsymbol{A}^\mathsf{H}$ denotes the conjugate transpose. For both ODER and RED (DEQ) during training, we fix the step-size parameter and regularization parameter to $\gamma = 5 \times 10^{-3}$ and $\tau = 4$, respectively. The learning rate of ODER/RED (DEQ) is set in two stages. In the first 100 epochs, we adopt the cyclic learning rate policy [19], where the policy cycles the learning rate between $0.05$ and $0.16$ with exponentially decay to $0.9998$. In stage 2, the learning rate was gradually reduced by a factor of $0.6$ every 50 epochs. The number of total training epochs was 200. We set the same forward pass initialization $\boldsymbol{x}^0$ in ODER for all reference methods. In these experiments, we set the number of forward pass iterations in ODER/RED (DEQ) to $K = 80$, and we set the steps in RED (Denoising) and RED (Unfold) to $K = 100$ and $K = 9$, respectively.

Table 2 reports average SSIM values obtained by ODER and other baselines. Fig. 2 presents quantitative evaluation of ODER on IDT for two minibatch sizes $w \in \{100, 250\}$ against RED (DEQ) using the full-batch of $b = 500$ measurements. Specifically, Fig. 2 (*left*) presents the empirical acceleration of ODER backward pass over that of RED (DEQ) due to the reduction in the computation complexity of data-consistency blocks at each iteration. Fig. 2 (*middle*) illustrates the loss against the epoch number on the training set, while Fig. 2 (*right*) presents the SNR achieved at different epochs for different values of $w$ evaluated over testing set. Fig. 3 provides additional visualizations of IDT reconstruction produced by ODER and other reference methods. Fig. 5 (left) shows the evolution of SNR for ODER, RED (DEQ), and RED (Denoising) on IDT against the iteration number.

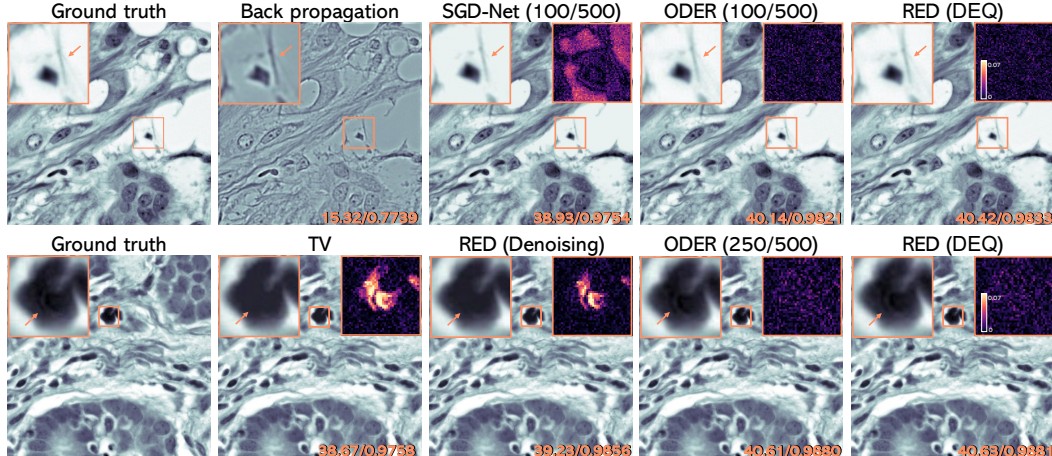

Figure 3: *Quantitative evaluation of several well-known methods on IDT under noise corresponding to input SNR of 20 dB. The total number of IDT measurements in this experiment is $b = 500$. RED (DEQ) corresponds to the full batch architecture that uses all the measurements at every step. Each image is labeled with its SNR (dB) and SSIM values with respect to the original image. The yellow box provides a close-up with a corresponding error map provided on its right. Note the similar performance of ODER and RED (DEQ), and the improvement over RED (Denoising) /RED (Unfold) due to the usage of DEQ learning. Best viewed on a digital display.*

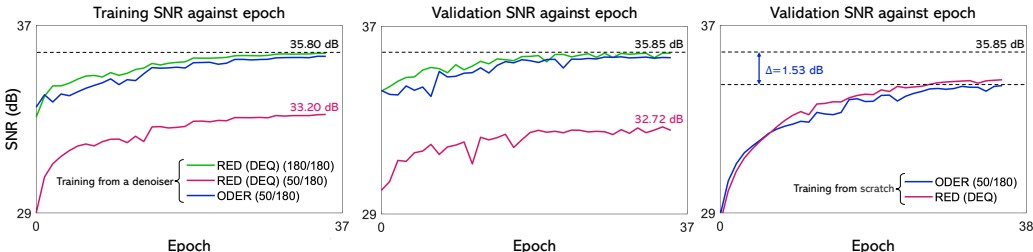

Figure 4: ***Left and Middle:*** *Average SNR is plotted against the training epoch for ODER and RED (DEQ) on the training and validation images used in the sparse-view CT experiments of the main paper. Two variants of RED (DEQ) are trained with $b = 50$ and $b = 180$ projections to illustrate the influence of using all the available measurements. While the complexity of ODER, which cycles through $w = 50$ projections, is comparable to RED (DEQ) using $b = 50$, it nearly achieves the performance of RED (DEQ) using all $b = 180$ projections.* ***Right:*** *ODER and RED (DEQ) in Left and Middle were trained by initializing the CNN prior using pre-trained denoisers. Here, we show the evolution of SNR for ODER and RED (DEQ) trained from a random initialization ("training from scratch"). Note the empirical benefit of initializing the CNN priors using pre-trained denoisers.*

## D.2 Additional Details and Validations for CT/MRI

**Sparse-view CT.** For the CT images, we train ODER by using the filtered backprojection (FBP) initialization $x^0 = A^H F y$. We use the Hann filter for FBP reconstruction. We set the number of forward pass steps in ODER/RED (DEQ) to $K = 180$, and we use Adam with training minibatch size 4 and weight decay $1 \times 10^{-7}$. For ODER and RED (DEQ), we fixed the step-size to $\gamma = 1.25 \times 10^{-3}$ and regularization parameter to $\tau = 3$. The learning rate starts from $3 \times 10^{-4}$ and is halved at epoch 15, then gradually reduced by a factor of 0.6 every 5 epochs. The number of total training epochs is 35. It is worth to note that we equally divided the full projection views $b \in \{90, 120, 180\}$ into 5 non-overlapping chunks, each with size of $\{18, 24, 36\}$ views, respectively. At every iterations, the corresponding ODER model with $w/b \in \{30/90, 40/120, 50/180\}$ randomly picks a subset of $\{6, 8, 10\}$ from each chunk for the data-consistency block calculation. This leads to better empirical

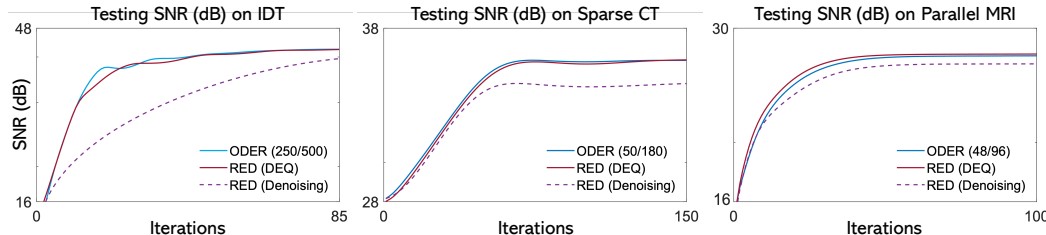

Figure 5: *Illustration of the SNR convergence of ODER, RED (DEQ) and RED (Denoising) for three imaging applications.* **Left:** *IDT with the full batch of $b = 500$ measurements under 25 dB input SNR.* **Middle:** *Sparse-view CT with $b = 180$ projection views.* **Right:** *Parallel MRI at $20\%$ sampling with $b = 96$ simulated coil sensitivity maps. Note that ODER matches the performance of RED (DEQ), and converges to a higher SNR than RED (Denoising) due to the end-to-end training.*

Table 3: Sparse-view CT reconstruction in terms of SNR (dB) and SSIM on another patient from [20].

| Method | Projection Views | | | | | |
|---|---|---|---|---|---|---|
| | 90 | | 120 | | 180 | |
| **U-Net** | 33.66 | 0.9775 | 34.62 | 0.9808 | 35.77 | 0.9848 |
| **RED (Unfold)** | 34.27 | 0.9810 | 35.18 | 0.9842 | 36.27 | **0.9870** |
| **RED (Denoising)** | 32.94 | 0.9717 | 34.07 | 0.9781 | 35.34 | 0.9812 |
| **ODER** | 34.69 | 0.9827 | 35.46 | 0.9848 | 36.21 | 0.9864 |
| **RED (DEQ)** | **34.81** | **0.9831** | **35.55** | **0.9852** | **36.34** | 0.9867 |

reconstruction improvement for ODER. Similarly, we set the number of forward pass iterations in RED (Denoising) and RED (Unfold) to $K = 150$ and $K = 7$, respectively.

Table 3 reports the average SNR for serveral methods on CT images using another different patient data from [20]. Table 4 presents new simulations on sparse-view CT to empirically quantify the influence of $w \geq 1$ on SNR (dB). Fig. 4 (left and middle) compares the average reconstruction SNR of ODER and RED (DEQ) for a fixed periteration measurement budget on sperase-view CT. The total number of projection views is $b = 180$. The batch algorithm RED (DEQ 50/180) and ODER with ($w = 50$) are allowed to use only 50 projection views at per iterate. This means that in each figure both RED (DEQ 50/180) and ODER with ($w = 50$) have the same per-iteration computational complexity. Empirically, we observe that ODER outperforms RED (DEQ 50/180) by around 3 dB on the validation set under the same per-step memory complexity and matches the performance of RED (DEQ 180/180). Fig. 4 (right) shows the average SNR of ODER and RED (DEQ) trained from a random initialization ("training from scratch") on CT. Note the empirical benefit of initializing the CNN priors using pre-trained denoisers. Fig. 5 (middle) shows the evolution of SNR for ODER, RED (DEQ), and RED (Denoising) on CT against the iteration number. Fig. 6 shows visual results of running ODER with different minibatch sizes $w \geq 1$ on CT test images, where the full measurement size is $b = 120$. Note how using $1/12th$ and $1/6th$ of total measurements allows ODER to match the visual quality of RED (DEQ). In Fig. 7 (*top*), we provide additional visualizations of the solutions produced by ODER/RED (DEQ) and various baseline methods considered in our work.

**Parallel MRI.** For the MRI images, we train ODER/RED (DEQ) by using the zero-filled reconstruction as initialization $x^0$. For the 2D MRI images, we set the number of iterations in ODER/RED (DEQ) to $K = 200$, and we use Adam with training minibatch size 16 and weight decay $2 \times 10^{-7}$. We fixed the step-size to $\gamma = 1.2$ and regularization parameter to $\tau = 0.05$ for both ODER and RED (DEQ). The learning rate starts from $1 \times 10^{-4}$ and is gradually reduced by a factor of 0.6 every 10 epochs. The number of total training epochs is 40. In these experiments, we set the number of forward pass iterations in RED (Denoising) and RED (Unfold) to $K = 200$ and $K = 25$, respectively. For the 3D MRI volumes, we set the number of steps in ODER/RED (DEQ) to $K = 400$, and we use Adam with training minibatch size 4 and weight decay $1 \times 10^{-6}$. We fixed the step-size to $\gamma = 1.25$ and regularization parameter to $\tau = 0.01$ for both ODER and RED (DEQ). The learning rate starts from $5 \times 10^{-5}$ and is gradually reduced by a factor of 0.65 every 10 epochs. The number of total training epochs is 100. Similar to sparse-view CT, we equally divided the number of coil sensitive maps

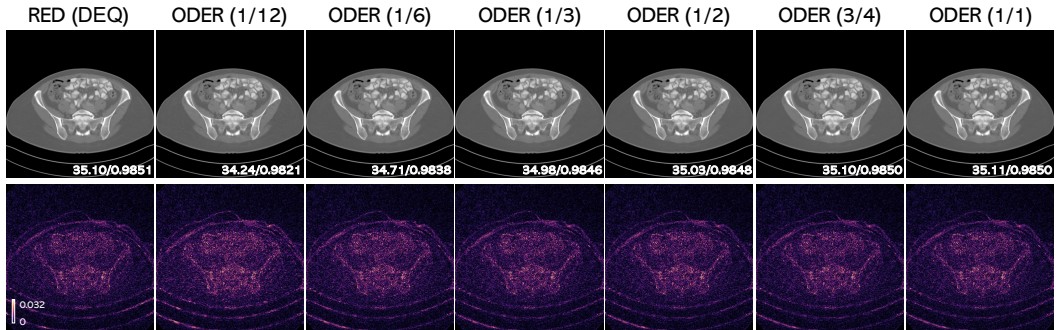

Figure 6: *Visual illustration of ODER results using different minibatch sizes $w \geq 1$ on CT test images. Each image is labeled with the corresponding SNR (dB) and SSIM values. The figures below are the error residuals relative to the ground truth. This figure corroborates our theoretical analysis by showing that $w$ allows to balance computational/memory efficiency against accuracy relative to RED (DEQ). Note how using 1/2th, 1/6th, or 1/12th of total measurements allows ODER to be within 0.07 dB, 0.39 dB, and 0.86 dB of SNR achieved by RED (DEQ) that uses all $b = 120$ measurements.*

Table 4: Average SNR (dB) of ODER for different minibatch sizes $w \geq 1$ on CT test images. Note that RED (DEQ) always uses the full batch size of $b = 120$ measurements.

| w/b, b=120 | 1/12 | 1/6 | 1/3 | 1/2 | 3/4 | 1/1 | RED (DEQ) |
|---|---|---|---|---|---|---|---|
| ODER | 34.34 | 34.94 | 35.12 | 35.24 | 35.25 | 35.27 | 35.26 |

$b = 96$ into 4 non-overlapping chunks, each with size of 24 coils, respectively. At every iterations, the corresponding ODER model with $w = 48$ randomly picks a subset of 12 from each chunk for the data-consistency block calculation. For the 3D MRI volumes, we set the number of forward pass iterations in RED (Denoising) and RED (Unfold) to $K = 400$ and $K = 20$, respectively. Fig. 5 (right) shows the evolution of SNR for ODER, RED (DEQ), and RED (Denoising) on MRI against the iteration number. Fig. 7 (*bottom*) reports the comparison on medical brain images for CS-MRI with under-sampling ratio of 10%.

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

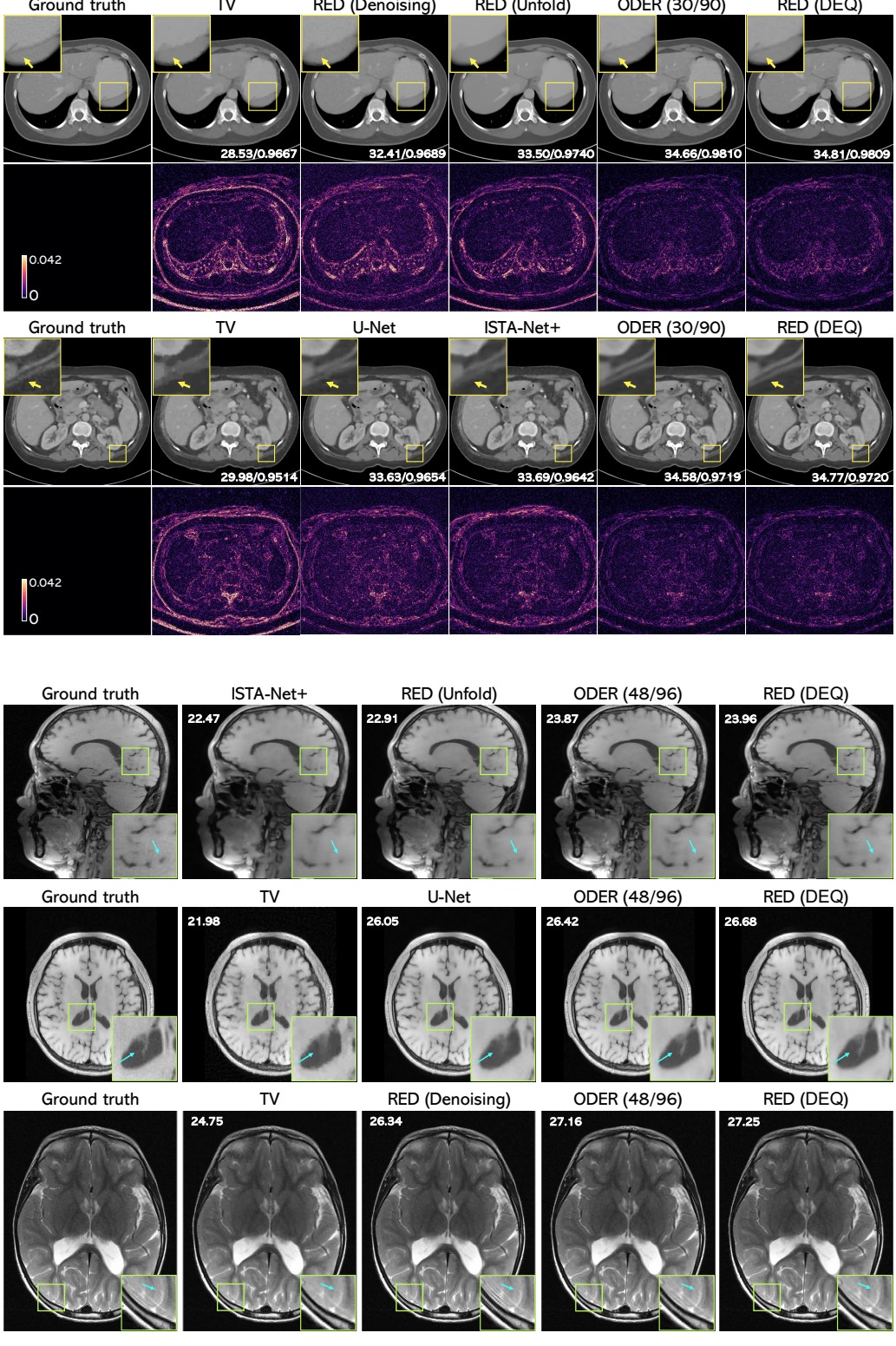

Figure 7: *Visual evaluation of several well-known methods on two imaging problems: (top) Reconstruction of sparse-view CT from $b = 90$ projection views. Each image is labeled with the corresponding SNR (dB) and SSIM values. The figures below are the error residual images to the ground truth; (bottom) Reconstruction of brain MRI images from its radial Fourier measurements at 10% sampling with $b = 96$ simulated coil sensitivity maps. Best viewed on a digital display.*