# OpenReview forum: "Online Deep Equilibrium Learning for Regularization by Denoising"
_NeurIPS.cc/2022/Conference — NeurIPS 2022 Accept_

### Official Review · Reviewer_2uPr · 2022-07-07

**Rating:** 5
**Confidence:** 4
**Soundness:** 3 good
**Presentation:** 3 good
**Contribution:** 2 fair

**Summary:**

In this paper, to address the issue that the training of Deep Equilibrium Models (DEQ) can still be a significant computational and memory challenge in applications that require processing a large number of sensor measurements, the authors propose Online Deep Equilibrium Learning for Regularization by Denoising (ODER) for inverse problems that adopts stochastic processing of measurements within an implicit neural network. Experiments on three applications demonstrate the potential improvements in training/testing complexity.

**Questions:**

1) The contribution needs to be better justified.
2) This work could explore to further improve the analysis and design variants of ODER to enhance its performance, as the authors also mentioned in Line 287, Page 9.
3) There are a few grammar errors. For instance, the section titles "4 Theoretical Analys" and "7 Broader impact" should be "4 Theoretical Analysis" and "7 Broader Impact".


**Ethics Review Area:**

["I don’t know"]

**Limitations:**

Yes

**Strengths And Weaknesses:**

This paper solves data-intensive imaging inverse problems and owns some strengths:
1) The main advantages of the presented technique are the reduced execution time w.r.t. the main baseline RED (DEQ) and the lower memory consumption of the measurement.
2) The paper is overall well structured and written.
3) The provided references are comprehensive and adequate.
4) The work seems correct since the proposed approach lays its foundations on robust literature papers, and the authors theoretically analyse ODER regarding its convergence and ability to approximate the traditional DEQ approach.

However, there are some issues that I would like to highlight:
1) Novelty/Originality/Contribution:
Although integrating the online processing into the DEQ framework is overall a novel attempt for solving inverse imaging problems using PnP/RED operators, it is a straight borrow from other literature and stitching.
I worry that the contribution of the paper is limited since the only improvement to DEQ and Ref.[35] is the addition of online processing. This seems like a small and straightforward contribution to pre-existing works and hardly motivates the publication of a new paper.
2) Significance:
This paper specifically addresses issues in imaging applications that require processing a large number of sensor measurements, which may limit its impact.
3) Experiment:
In Figure 3, Tables 2 and 3, the SSIM and SNR of the proposed approach are not better than RED (DEQ) in experiments on CT and MRI images, so the only real contribution is the reduced execution time.

---

> ### Author Response · Authors · 2022-08-02
> **Response to Reviewer 2uPr**
>
> Thank you for your valuable feedback. As discussed below, the feedback has prompted us to adjust the presentation of our work to better reflect its contributions.
>
> **Weakness:**
> >**1:**
> * We understood from the feedback that the presentation of our work did a disservice to its contributions. It is worth highlighting that (**a**) _learning implicit online neural networks_ has never been considered before our work and (**b**) such learning can enable nearly 2.5x improvement in training time (see Fig. 1 in the main paper). Therefore, we would argue that ODER is not just an addition of online processing, but a theoretically sound method that leads to substantial practical improvements in DEQ for inverse problems. Theorem 3 in the initial submission establishes bounds on the training accuracy of implicit online neural networks, which is different from prior work proving convergence of traditional online algorithms. It is a nontrivial theoretical result that online forward and backward iterations in ODER can provide gradients with controllable accuracy for training implicit online neural networks. We are not aware of any prior work that has considered similar ideas or performed similar theoretical analysis.
> - Based on the feedback, we realized that we can improve the presentation of our work by renaming Theorem 3 to be the Main Theorem. Theorems 1 and 2 are renamed Propositions 1 and 2 to highlight their supporting roles. Similarly, we move Section 3.3 to be Section 3.1, since this is where the main innovation of ODER as a method for learning implicit online networks is presented (see the revised paper).
>
> >**2:**
> - We were surprised by the reviewer’s comment that the focus on imaging applications that require processing a large number of sensor measurements may limit the impact of our work. On the contrary, we can list examples of imaging applications that have historically relied on efficient processing of a large number of sensor measurements, such as the three applications considered in the paper (sparse-view CT and parallel MRI in medicine, and IDT in bio-microscopy), as well as other well-known applications, including  photoacoustic tomography (PAT), positron emission tomography (PET), optical projection tomography (OPT), and optical diffusion tomography (ODT). All those applications are widely used, and would potentially benefit from powerful and efficient deep model-based architectures such as ODER.
>
> >**3:**
> * Indeed, a practical conclusion of our work is that ODER can reduce the training time of RED (DEQ) for the same SNR/SSIM performance. Figure 1 shows that ODER reduces the training time by 2.5 times for the same SNR, which is important for real applications.
> - Our work also presents novel theoretical analysis in Theorem 3 of the original submission that does not exist in prior work. Our main theorem shows that implicit online networks can approximate DEQ to a desired precision _during training_, which is different from existing work on inverse problems that shows convergence _during inference_. Note that it is a nontrivial theoretical result that randomized forward and backward iterations end up enabling the computation of the training gradients with controllable accuracy.
>
> **Questions:**
> >**1:**
> - As discussed above, we renamed Theorem 3 to Main Theorem to highlight its novelty compared to the existing work. We also highlighted the novelty of training implicit online networks by moving Section 3.3 to Section 3.1 in the revision. The abstract, introduction, and conclusion were revised accordingly.
>
> >**2:**
> - Our paper presents a novel algorithm and mathematical analysis that has never been considered in the existing work in the area. Inclusion of additional results would distract from our main message, which motivates us to leave it for future work.
>
> >**3:**
> - Thank you! We will correct these typos in the revised paper.

---

> > ### Comment · Reviewer_2uPr · 2022-08-07
> > **Post-rebuttal: Some concerns are addressed**
> >
> > I would like to thank the authors for their effort to provide response in explaining the contributions. It is clarified now that the theoretical analysis is important. Thus, I raise my score by one.
> >
> > Although this work has some theoretical contribution, I still believe that this work has limited practical applications . Only reducing the training time is not that significant.

---

> > > ### Author Response · Authors · 2022-08-09
> > > **Additional comments for Reviewer 2uPr**
> > >
> > > First of all, thank you for reviewing our responses, raising your score, and your valuable feedback.
> > >
> > > With the hope of further increasing your evaluation, let us highlight another benefit of ODER that goes beyond training time reduction. ODER is useful for improving the memory complexity of training, which is crucial when GPU memory is the bottleneck. Note that when the GPU memory is the bottleneck, the performance of the traditional DEQ training will drop (see Figure 4 in the supplementary material). As can be seen in Table 1 of the main paper, ODER can reduce the GPU usage from 3.56 GB to 0.71 GB, which is about 80% reduction. This reduction essentially comes for free, since ODER matches the performance of the traditional DEQ training.

---

### Official Review · Reviewer_uDGc · 2022-07-11

**Rating:** 6
**Confidence:** 5
**Soundness:** 3 good
**Presentation:** 3 good
**Contribution:** 2 fair

**Summary:**

This paper introduces an online deep equilibrium learning approach for large-scale inverse imaging problems. The method is built upon the recently proposed deep equilibrium architecture that unrolled an optimization algorithm (e.g., steepest gradient decent for regularization by denoising) into a neural network with a potential infinite number of depth (iterations). This paper goes beyond that by presenting an online variant to enable large-scale inverse imaging applications where the full evaluation of data consistency term is quite expensive. It therefore demonstrates superior performance on three data-intensive inverse imaging tasks -- its reconstruction quality is comparable to the full batch solution while is 2-3x faster, suggesting the benefits of the proposed method

**Questions:**

I'd like to see the mean iteration times of the deep equilibrium architecture. Does end-to-end training help to reduce the number of iterations for convergence, compared with the PnP/RED counterpart?

Recent progresses on PnP would also be worthy to be mentioned, such as [D, E, F]

[D] It Has Potential: Gradient-Driven Denoisers for Convergent Solutions to Inverse Problems, NeurIPS 2021
[E] TFPnP: Tuning-free Plug-and-Play Proximal Algorithms with Applications to Inverse Imaging Problems, JMLR 2022
[F] Gradient Step Denoiser for convergent Plug-and-Play, ICLR 2022

**Strengths And Weaknesses:**

The major strength of this work lies at the strong empirical evidence on three practical large-scale inverse imaging problems. Plus, the paper is nicely written and easy to follow. The theoretical convergence of the algorithm is also rigorously analyzed.

Nevertheless, the technical novelty of this work is rather incremental. The methodology here is mostly credited to [A], which first connected the deep equilibrium model and PnP/RED method. The nontrivial technical parts thus can only be attributed to the online version, as well as theoretical analysis of convergence, but none of them could be viewed as significant contributions unfortunately. Virtually, the deltas here are well known in the PnP literatures, e.g., online version of PnP and RED algorithms are proposed in [B] and [C] respectively, even both of them have already established the theoretical convergence.

Consequently, given the pros and cons on balance, I feel this is a very borderline paper, and I vote for borderline accept tentatively
***
**I raise my score because of the originality of the main theory highlighted in the rebuttal.**

[A] Deep Equilibrium Architectures for Inverse Problems in Imaging, TCI 2021
[B] An Online Plug-and-Play Algorithm for Regularized Image Reconstruction, TCI 2019
[C] Block Coordinate Regularization by Denoising, NeurIPS 2019

---

> ### Author Response · Authors · 2022-08-02
> **Response to Reviewer uDGc**
>
> Thank you for your valuable feedback. As discussed below, the feedback has prompted us to adjust the presentation of our work to better reflect its contributions.
>
> **Weakness:**
>
> - We were initially surprised by the reviewer’s evaluation that our work is similar to the prior work in [A, B, C]. However, upon careful re-reading of our initial submission, we realized that this confusion can be attributed to the way we presented our results. Based on the feedback, we can improve the presentation by renaming Theorem 3 to be the Main Theorem. Theorems 1 and 2 are renamed Propositions 1 and 2 to highlight their supporting roles. Similarly, Section 3.3 will move to become Sections 3.1, since this is where the main innovation of ODER as a method for learning implicit online networks is proposed (see the revised paper).
> * ODER is conceptually different from the traditional online PnP/RED algorithms for solving inverse problems [B, C]. The nuance is that ODER is a framework for _training implicit online neural networks_, which is different from _running online PnP/RED algorithms_ at inference time. This topic has never been considered in the existing literature. Additionally, our analysis in Theorem 3 in the initial submission is completely new as it establishes bounds on the training accuracy of implicit online networks, which is different from prior work [A, B, C] that proves convergence at inference time. It is a nontrivial theoretical result that online forward and backward iterations in ODER can provide gradients with controllable accuracy for training implicit online neural networks. We are not aware of any prior work that has proposed similar ideas or analyses as in our paper.
>
> **Questions:**
>
> >**1:**
>
> | Method          | Sparse-view CT  | Parallel MRI |
> |-----------------|:---------------:|:------------:|
> | ODER            |        43       |      72      |
> | RED (DEQ)       |        35       |      43      |
> | RED (Denoising) |        30       |      45      |
>
> * Prompted by your comment, we prepared the table above that reports the mean iteration numbers for ODER, RED (DEQ), and RED (Denoising), for the same stopping tolerance, in sparse-view CT and parallel MRI. Note how end-to-end training of implicit networks—ODER and RED (DEQ)—can increase the mean iteration numbers compared to the traditional RED (Denoising). This is reasonable since the implicit networks are trained to maximize the SNR, not to minimize the number of iterations. By allowing for more iterations, implicit networks can get trained to achieve higher SNR than RED (Denoising) (≥ 1 dB for CT in Table 2 and ≥ 0.5 dB for MRI in Table 3 of the main paper).
>
> >**2:**
>
> - Our original submission has unintentionally overlooked the references [D-F], which we added in the revised version (Refs [7, 66, 74] in the revised paper). The proposed ODER training and its analysis are perfectly compatible with the iterations based on the variations of PnP in [D-F].

---

> > ### Comment · Reviewer_uDGc · 2022-08-08
> > **The rebuttal addresses my concerns**
> >
> > Though I still feel the contribution of training implicit online network is moderate given the existing work [A], I do agree with the authors the main theory presented in this work is entirely new and interesting. As a result, I raise my score to Weak Accept.
> >
> > The following questions would not impact my final rating, but I'm curious is the trained proximal mapping still a valid denoiser for Gaussian denoising, after end-to-end training? What about directly plugging the learned "denoiser" (on CS-MRI) into another task (sparse-view CT)?

---

> > > ### Author Response · Authors · 2022-08-09
> > > **Additional comments for Reviewer uDGc**
> > >
> > > | Method    |     UNet     | RED (Unfold) |     ODER     |   RED (DEQ)  |
> > > |-----------|:------------:|:------------:|:------------:|:------------:|
> > > | PSNR/SSIM | 18.86/0.8824 | 20.55/0.9215 | 23.47/0.9441 | 23.48/0.9450 |
> > >
> > >
> > > First of all, thank you for reviewing our responses, raising your score, and your valuable feedback.
> > >
> > > With the hope of further increasing your evaluation, let us highlight that finding ways to efficiently train implicit neural networks is an emerging area with a big potential for practical impact. While [A] proposes to use implicit networks for inverse problems, it does not consider the impact of the forward model on the memory and computational complexity of training. This is something our work addresses and does so with new a new method, theoretical insights, and numerical results not available in [A].
> > >
> > > We ran simulations to, at least partially, address your second comment. The table above provides PSNR/SSIM results for sparse-view CT with b=120 for ODER, RED (DEQ), RED (Unfold), and UNet trained on CT images, but tested on 50 Brain MRI images. It is worth mentioning that the CNN priors in RED (Unfold), ODER, and RED (DEQ) are not proximal operators, since they are not trained to be denoisers. They can be better viewed as artifact removing operators trained for end-to-end optimality. Most importantly, note how the relative performance of ODER matches that of RED (DEQ) even for mismatched CNN priors trained on CT but tested on MRI. Fully addressing your comment by including a forward model mismatch would require a more careful evaluation, which would extend beyond this review window.

---

### Official Review · Reviewer_a7f5 · 2022-07-13

**Rating:** 7
**Confidence:** 4
**Soundness:** 3 good
**Presentation:** 4 excellent
**Contribution:** 3 good

**Summary:**

Edit - score raised by 1 point based on author revisions.

This work proposes ODER, an online learning method for deep equilibrium learning for Regularization by Denoising. DEQ is a recently proposed framework for memory efficient learning of an infinite-depth unrolled network as implicitly defined by a fixed point of an operator. RED is a specific type of iterative algorithm that can incorporate learned priors, and has the fixed point operator structure. The authors therefore apply stochastic gradient descent across the measurement direction for each input in the training set, and derive the corresponding update equations for gradient-based learning. They show that their approach is able to get similar quality reconstructions with reduced memory and training time, compared to full-fledged DEQ-RED.


**Questions:**

- Could the CT experiments be run with a larger test set or with cross-validation?

- How sensitive is the method to changes in the measurement operator between train and test?


**Limitations:**

I don't think that Section 6 sufficiently describes limitations of the work. Could the authors discuss limitations of the approach, for example the required theoretical assumptions, implementation considerations, etc.? Another example is the relative improvement of ODER over RED (Unfold) or RED (Denoising). As both of these alterntives can also be implemented with online learning, how will the training time compare?


**Strengths And Weaknesses:**

Strengths:
The paper is well-organized and presentation is clear. The work nicely connects online learning with DEQ, with specific application to RED. There is a "free-lunch" result: both memory and time to train are reduced, without any sacrifice in image quality. For this reason this work is exciting. The work is also supported by theoretical results and is demonstrated for a several different applications.

Weakness:
As the authors state, online-learning for RED is not a new concept, and both are special cases of running SGD across the measurement direction (e.g. coils in MRI) which is not novel on its own; for example:
[1] Ong, F, Zhu, X, Cheng, JY, et al. Extreme MRI: Large-scale volumetric dynamic imaging from continuous non-gated acquisitions. Magn Reson Med. 2020; 84: 1763– 1780. https://doi.org/10.1002/mrm.28235

The CT experiment had only one subject in the test set, which seems prone to overfitting.

---

> ### Author Response · Authors · 2022-08-02
> **Response to Reviewer a7f5**
>
> Thank you for the feedback and positive comments on our work. Please see below for our responses.
>
> **Weakness:**
> * ODER is different from the traditional online learning algorithms (SGD, Adam) and from the traditional online algorithms for solving inverse problems (online PnP/RED or Ref [1] by the reviewer). ODER is for training _implicit online neural networks_, which has never been explored in the existing literature. Theorem 3 in the initial submission (Main Theorem in the revision) establishes bounds on the training accuracy of implicit online networks, which is different from prior work proving convergence of online algorithms. It is a nontrivial theoretical result that online forward and backward iterations in ODER can provide gradients with controllable accuracy for training implicit online neural networks. We are not aware of any prior work that has explored similar ideas or analysis.
> - We can improve the presentation of our work by renaming Theorem 3 to be Main Theorem. Theorems 1 and 2 are renamed Propositions 1 and 2 to highlight their supporting roles for the Main Theorem. Similarly, we move Section 3.3 to be Section 3.1, since this is where ODER is presented as a method for learning implicit online neural networks.
> * We cited Ref [1] mentioned by the reviewer in the context of online methods for solving imaging inverse problems (Ref [48] in this revision).
> - Please see our response to Question 1 below.
>
> **Questions:**
> >**1:**
>
> | Method \ # Projections |90|120|180|
> |-|:-:|:-:|:-:|
> | ODER|34.69/0.9827|35.46/0.9848|36.21/0.9864|
> | RED (DEQ)|34.81/0.9831|35.55/0.9852|36.34/0.9867|
> | RED (Denoising)|32.94/0.9717|34.07/0.9781|35.34/0.9812|
> | RED (Unfold|34.27/0.9810|35.18/0.9842|36.27/0.9870|
>
> - Prompted by your comment, we did additional tests using 55 slices of size 512 x 512 from another patient, who was not part of our original data. The table above presents corresponding SNR and SSIM values for ODER, RED (DEQ), RED (Unfold), and RED (Denoising). Note the consistency of these results with those presented in Table 2 of the main paper. We are confident that the existing numerical results on ODER are representative of its general behavior. If requested by the reviewers, we will be happy to include additional test samples for computing the results in Table 2, although we do not expect that those will impact our contributions.
> * We will release our implementation of ODER on all the considered modalities.
> - The numerical results on CT in our original submission used a public dataset from a CT challenge (Ref [87] in the revised paper). We adopted a similar setting used in the literature cited in the main paper (Refs [11, 12, 14]).
>
> >**2:**
>
> | # Projections (b)|90|120|180|
> |-|-|-|-|
> |ODER (trained using b=120)|34.37|35.12|35.81|
> |ODER (trained using a matched b)|34.40|35.12|35.91|
> |RED (DEQ) (trained using b=120)|34.41|35.26|35.95|
> |RED (DEQ) (matchet b)|34.61|35.26|35.95|
>
> - We did additional tests on CT using a mismatched number of measurements in training and testing. The table above shows results for two variants of ODER: one trained using 120 measurements and the other trained using the same number of measurements as for testing. We also provide RED (DEQ) for reference. Note how the performance of ODER trained using b = 120 is always within 0.1 dB of the oracle ODER that has a perfectly matched training and testing b.
>
> |Input SNR (dB)|50|45|40|35|
> |-|-|-|-|-|
> |ODER|35.12 |33.11|30.19|24.21|
> |RED (DEQ)|35.26|33.20|30.28|24.48|
> |RED (Unfold)|35.01|32.90|28.87|23.75|
>
> - We also ran tests on CT using a mismatched amount of Gaussian noise. The table above shows the performance of three methods—ODER, RED (DEQ), and RED (Unfold)—trained at 50 dB input SNR and tested at 35, 40, 45, and 50 dB input SNR.
> * Based on the new results, we can conclude that ODER is relatively robust to shifts in the number of measurements and the amount of noise, in the sense that it maintains its relative reconstruction performance with respect to RED (DEQ).
>
> **Limitations:**
> - If the reviewer requests, we can revise our conclusion to include the following: (a) ODER has an additional hyperparameter w ≥ 1 to control accuracy/efficiency tradeoff relative to RED (DEQ); (b) just like RED (DEQ), ODER training achieves memory savings over RED (Unfold) at the cost of higher computational complexity due the computation of the forward and backward fixed points; (c) just like RED (DEQ) and RED (Unfold), ODER recovers better images over RED (Denoising) at the cost of an overall more expensive training using the measurement model.
> * Note that while RED (Denoising) can be implemented as an online inference algorithm, it is different from ODER since it doesn’t learn the prior end-to-end with the measurement model. On the other hand, while RED (Unfold) can also be implemented using online learning, it is not an implicit network and thus suffers from high memory complexity from storing intermediate latent variables across unfolded iterations.

---

> > ### Comment · Reviewer_a7f5 · 2022-08-08
> > **Concerns are addressed**
> >
> > Thank you for the detailed response to my comments. Though the work is incremental, I believe it has potential to be used for many applications. I have raised my score to reflect the revisions.

---

> > > ### Author Response · Authors · 2022-08-09
> > > **Acknowledgement**
> > >
> > > Thank you for reviewing our responses, raising your score, and your valuable feedback.

---

### Official Review · Reviewer_Hjnn · 2022-07-14

**Rating:** 6
**Confidence:** 5
**Soundness:** 3 good
**Presentation:** 3 good
**Contribution:** 3 good

**Summary:**

The paper studies an online Deep Equilibrium Models (DEQ) method for Regularization by Denoising (RED). The proposed ODER incorporated randomized processing of measurements. The ODER algorithm aims to bypass the high computational/memory complexity of DEQ led by the high dimensionality of measurement space. The introduced online backward pass is demonstrated to lead to a more scalable and flexible DEQ framework for inverse problems. Based on standard assumptions in the analysis of fixed-point and SGD, the authors also have given some analysis of the ODER's convergence. The authors have also conducted experiments to analyze the behaviour of ODER, and demonstrate its effectiveness.  As an accelerated DEQ-RED, the proposed ODER is a useful extension to DEQ and RED.





**Questions:**

1. Line 122. I am curious if any w<<b enough to make ODER work? Any empirical/theoretical lower bound for w to guarantee ODER work? What are the principles for choosing w in practice?

2. Line 210. "The CNN prior of ODER is initialized using pre-trained denoisers .." What's the pretraining strategy? What if train the CNN prior from scratch?

3. Line 220 - 221. Do you mean one needs to train 5 denoisers for different noise levels? If so, how bad if we train a denoiser to model all noise levels?

4. line 220. I am confused on if only the AWGN denoisers are valid to use? Is the ODER also scalable to other noise types? How to construct/select denoisers? What if BM3D or DnCNN are used in ODER?



**Ethics Review Area:**

["I don’t know"]

**Limitations:**

Suggestions: there are other relevant works that are worth mentioning. As far as I know, in addition to the denoiser prior, the below DL paradigms also incorporate the knowledge of data acquisition to solve inverse imaging problems which have been studied for the purpose of learning Generative prior [1], Neumann inversion [2] or Equivariance prior [3]. To a broader view, it's necessary to add some review or comparison with these related paradigms.

[1] Bora, A., Price, E., & Dimakis, A. G. (2018). AmbientGAN: Generative models from lossy measurements. In International conference on learning representations.

[2] Gilton, D., Ongie, G., & Willett, R. (2019). Neumann networks for linear inverse problems in imaging. IEEE Transactions on Computational Imaging, 6, 328-343.

[3] Chen, D., Tachella, J., & Davies, M. E. (2021). Equivariant imaging: Learning beyond the range space. In Proceedings of the IEEE/CVF International Conference on Computer Vision (pp. 4379-4388).

**Strengths And Weaknesses:**

Strengths:

1. The studied problem is interesting and worth devoting to, and the author has done significant work.
2. The analysis and experiments are extensive and can support the claims.
3. The paper is well written and easy to follow.

Weakness:

1. 'Online processing of measurements' and its analysis tricks are not new.
2. Some experimental settings and studies on hyper-parameters are not well discussed (see the points in [Questions]).

My initial rating of the paper is weak accept. I am open to raising my score if the authors can address my concern during the rebuttal.

---

> ### Author Response · Authors · 2022-08-01
> **Response to Reviewer Hjnn**
>
> Thank you for the feedback and positive comments on our work. We provide point-by-point responses to your comments below.
>
> **Weakness:**
> > **1:**
> * We would like to point out that Theorem 3 in our original submission is novel and the learning technique in Section 3.3 of our original submission is also novel in the context of “online processing of measurements.” The novelty of the proposed method is in _training_ of implicit online networks, which is different from existing work on running online PnP/RED using pre-trained denoisers as priors. The novelty of the main theorem is in the bound for the training accuracy of implicit online neural networks, which is different from existing work proving convergence of PnP/RED. It is a nontrivial theoretical result that online forward and backward iterations in ODER can provide gradients with controllable accuracy for training implicit online networks. We are not aware of any prior work that has explored similar ideas or analysis.
> - Based on the feedback, we realized that we can improve the presentation of our work by renaming Theorem 3 to be Main Theorem. Theorems 1 and 2 are renamed Propositions 1 and 2 to highlight their supporting roles for the Main Theorem. Similarly, we moved Section 3.3 to be Sections 3.1 in the revision, since this is where ODER is presented as a method for learning implicit online neural networks.
>
> >**2:**
> * Please see below for our answers to the raised questions. All the additional simulations in our answers will be included in the revised supplementary material.
> - We will publicly release our code upon the acceptance of the paper, which will simplify the communication of the hyperparameters for the reproducibility of our work.
>
> **Questions:**
> >**1:**
>
>   |w/b, b=120|1/12|1/6|1/3|1/2|3/4|1|RED (DEQ)|
>   |:-|:-:|:-:|:-:|:-:|:-:|:-:|:-:|
>   |ODER|34.34|34.94|35.12|35.24|35.25|35.27|35.26|
>
> * Prompted by your remark, we ran new simulations on sparse-view CT to empirically quantify the influence of w ≥ 1 on SNR (dB). The table corroborates our theoretical analysis showing that the parameter w allows to balance computational/memory efficiency against accuracy relative to RED (DEQ). Note how using 1/2th, 1/6th, or 1/12th   of total measurements allows ODER to be within 0.02 dB, 0.4 dB, and 1 dB, respectively, of the SNR achieved by RED (DEQ) that uses all b = 120 measurements.
> - In practice, w is best treated as a hyperparameter whose value is empirically optimized for the best imaging performance (SNR, SSIM) under acceptable hardware constraints (GPU memory, training time). Our main theorem ensures that ODER can approximate RED (DEQ) to a desired accuracy for large enough w.
>
> >**2:**
>
> |Methods|Training from scratch|Training from a denoiser|
> |-|:-:|:-:|
> | ODER| 34.56| 35.91|
> | RED (DEQ)|34.73|35.95|
>
> - We adopted the pre-training strategy from the reference [40] in the revised main paper, which used pre-trained denoisers to initialize the traditional DEQ. Prompted by your remark, we ran new simulations on sparse-view CT to empirically quantify the benefit of initializing from a pre-trained denoiser. The table above compares the performance of ODER and RED (DEQ) trained from a random initialization (“Training from scratch”) against those trained from a pre-trained denoiser. Note the empirical benefit of initializing the CNN priors using pre-trained denoisers, which justifies our strategy of using pre-trained denoisers for initializing ODER.
>
> >**3:**
> - ODER doesn’t use denoisers as priors (beyond initialization, see our response to Question 2) since its purpose is to train implicit online networks. The CNN prior in ODER is trained for end-to-end performance.
> * The 5 denoisers are for the traditional RED (Denoising), not our proposed method (see Table 1 of the main paper). For RED (Denoising), we simply adopted the commonly used strategy of using multiple denoisers (see Refs [24, 40, 59, 63] in the revised paper).
>
> >**4:**
> - The denoiser mentioned in Line 220 is for the traditional RED (Denoising), which is simply one of the methods we compare against in Table 1 of the main paper.
> * ODER does not use denoisers as priors, since its purpose is to train implicit online networks by using the algorithm in Section 3.3 in the initial submission (now Section 3.1 in the revision). Using BM3D or any other pre-trained denoiser would defeat the purpose of ODER by turning it into a traditional online PnP/RED method.
> - Any CNN architecture can in principle be trained using ODER. We tested 3 architectures, DnCNN, U-Net, and Tiny U-Net, and finally selected Tiny U-Net for ODER due to its computational efficiency. The ODER backward pass using Tiny U-Net is about 3x faster compared to the one using DnCNN or U-Net (see Table 1 in the supplement).
>
> **Limitations:**
> * We thank the reviewer for suggesting additional references! It is our goal to have a broad and well presented view of the area, and we cited the suggested references in the revised paper (Refs [4,5,6]).

---

> > ### Author Response · Authors · 2022-08-09
> > **Author Response**
> >
> > Dear reviewer, thank you again for reading our original submission and providing valuable feedback. The Author-Reviewer discussion ends today. If there is something that you would like to discuss beyond our responses above, please let us know and we will be happy to do so.

---

### Author Response · Authors · 2022-08-01
**Official Response to all reviewers:**


* Thank you all for providing us with valuable feedback. We provide detailed answers to all the questions below. To better address some of them, we ran additional simulations. We have **_updated the paper and the supplementary material_** based on our responses below. It is worth mentioning that the raised points helped to streamline and improve the paper without fundamentally altering the main contributions of our work.

- We were initially surprised that the reviewers saw similarity between our work and the existing literature on online PnP/RED. However, upon carefully re-reading our initial submission, we realized that this confusion can be attributed to the way we presented our results. The confusion can be alleviated by highlighting that Theorem 3 in our original submission is our main theoretical contribution, while Theorems 1 and 2 serve primarily as supporting results. We are not aware of any result in the existing literature analogous to our Theorem 3, which shows that _implicit online neural networks_ can be _trained_ to approximate DEQ for inverse problems to a desired precision. This result is different from the existing work showing the convergence of traditional PnP/RED methods. **_It is a nontrivial theoretical result that online forward and backward iterations in ODER can provide gradients with controllable accuracy for training implicit online neural networks for solving imaging inverse problems._**

* ODER is a conceptually different method from PnP/RED. Note how ODER is for training implicit online neural networks, which is different from running PnP/RED using pre-trained denoisers as priors. We are not aware of any method similar to ODER in the prior work on PnP/RED and hope that reviewers will appreciate the nuances between ODER and PnP.

- To highlight our main theoretical contribution, the revised paper renames Theorem 1, Theorem 2, and Theorem 3, to Proposition 1, Proposition 2, and Main Theorem, respectively. To highlight our main algorithmic contribution on learning implicit online neural networks, the revised paper moves Section 3.3 to be Section 3.1. The rest of the paper has been edited to be consistent with this naming and order of presentation.

---

### Comment · Area_Chair_Kxo2 · 2022-08-07
**Discussion period**

Thank you to all the reviewers for the great effort in reviewing the paper and the authors for the responses.

As the author-reviewer discussion period is almost over, I want to ensure that reviewers have read the authors' responses and engage with the authors if needed.

If you haven't done this, could you please take a moment to read through the authors' responses, update the reviews to indicate that you have read the authors' responses, or communicate with the authors if needed? You can also share in private conversations with the reviewing team.

Please continue to share your thoughts. Thank you!

---

### Author Response · Authors · 2022-08-09
**Discussion Period Response to All Reviewers and Area Chairs**

Thank you all again for reviewing our work. An additional thanks to those reviewers that have already read our responses and the area chair for managing the review of our paper. Let us know if there is anything else we can do to improve your evaluation of our work.

---

### Meta-Review · Area_Chair_Kxo2 · 2022-08-21

**Recommendation:** Accept
**Confidence:** Certain

**Metareview:**

The paper proposes a learning method (specifically a deep equilibrium learning approach) for 'regularization by denoising', a plug-and-play method for solving inverse problems.

After the rebuttal, all reviewers support acceptance of the paper. The reviewers find the paper to be well written, the problem to be interesting, and the claims to be well supported (reviewer Hjnn), both empirically (reviewer uDGc) and through theory. Reviewer A7f5 finds the work particularly exciting since both memory and training time are reduced, without sacrificing image quality.

Based on my own reading and the unanimous support of the reviewers, I recommend acceptance of the paper. A nice contribution!


**Award:**

No

---

### Decision · Program_Chairs · 2022-09-14

Accept